# Two domains of Tim50 coordinate translocation of proteins across the two mitochondrial membranes

Marcel G Genge[1], Shalini Roy Chowdhury[1], Vít Dohnálek[2], Kaori Yunoki[3], Takashi Hirashima[3], Toshiya Endo[3], Pavel Doležal[2], Dejana Mokranjac[1]

Hundreds of mitochondrial proteins with N-terminal presequences are translocated across the outer and inner mitochondrial membranes via the TOM and TIM23 complexes, respectively. How translocation of proteins across two mitochondrial membranes is coordinated is largely unknown. Here, we show that the two domains of Tim50 in the intermembrane space, named core and PBD, both have essential roles in this process. Building upon the surprising observation that the two domains of Tim50 can complement each other *in trans*, we establish that the core domain contains the main presequence-binding site and serves as the main recruitment point to the TIM23 complex. On the other hand, the PBD plays, directly or indirectly, a critical role in cooperation of the TOM and TIM23 complexes and supports the receptor function of Tim50. Thus, the two domains of Tim50 both have essential but distinct roles and together coordinate translocation of proteins across two mitochondrial membranes.

## Introduction

Over half of eukaryotic proteins need to be translocated across at least one intracellular membrane to reach the final place of their function. Proteins residing in the mitochondrial matrix are synthesized on cytosolic ribosomes as precursor proteins with the N-terminal, positively-charged, cleavable extensions called presequences (Vögtle et al, 2009; Neupert, 2015; Schulz et al, 2015; Grevel et al, 2019; Hansen & Herrmann, 2019; Genge & Mokranjac, 2021; Araiso et al, 2022; Busch et al, 2023). They have a particularly complicated journey as they need to find the right organelle and then cross its two membranes, the outer (OM) and the inner membrane (IM), and the aqueous subcompartment between them, the intermembrane space (IMS), before they reach their final destination. Translocation of presequence-containing precursor proteins across two mitochondrial membranes, also termed the presequence pathway, is mediated by a concerted action of the TOM (translocase of the outer mitochondrial membrane) and the TIM23 complexes (translocase of the inner mitochondrial membrane 23) in the outer and inner membranes, respectively.

Presequences are initially recognized on the mitochondrial surface by the receptors of the TOM complex, mainly Tom20 and Tom22 (Brix et al, 1999; Abe et al, 2000; Yamano et al, 2008), and then cross the OM through the central protein-import channel formed by the β-barrel protein Tom40, which is surrounded by the three small Tom proteins, Tom5, Tom6, and Tom7 (Bausewein et al, 2017; Araiso et al, 2019; Tucker & Park, 2019; Wang et al, 2020; Su et al, 2022). Two such barrels are tethered to each other by the transmembrane (TM) segments of two Tom22 molecules. Presequences exit the TOM channel at the side where the two barrels come together and which extends to the *trans* site of the TOM complex formed by the IMS-exposed segments of Tom22, Tom40, and Tom7 (Kanamori et al, 1999; Esaki et al, 2004; Shiota et al, 2011; Araiso et al, 2019; Rapaport, 2019; Tucker & Park, 2019). The available structures reveal that the three proteins converge at the IMS side of the TOM complex, but, unfortunately, the *trans* site itself remains structurally unresolved. Already at this stage, the IMS-exposed receptor of the TIM23 complex, Tim50, likely together with the N-terminal segment of Tim23 and possibly Tim21, recognizes the presequences (Yamamoto et al, 2002; Mokranjac et al, 2003, 2009; Chacinska et al, 2005; Tamura et al, 2009; La Cruz et al, 2010; Marom et al, 2011; Lytovchenko et al, 2013). In a membrane potential-dependent step, presequences are then delivered to the core of the TIM23 complex, formed by the IM-embedded segments of Tim23 and Tim17, which provides passage across the IM (Truscott et al, 2001; Martinez-Caballero et al, 2007; van der Laan et al, 2007; Alder et al, 2008a; Malhotra et al, 2013; Demishtein-Zohary et al, 2015, 2017; Ramesh et al, 2016; Wrobel et al, 2016; Fielden et al, 2023; Im Sim et al, 2023). Complete translocation of precursor proteins into the matrix requires multiple ATP-dependent cycles of the import motor of the TIM23 complex. Tim44 recruits the import motor, the ATP-consuming chaperone mtHsp70 (Ssc1 in yeast), and its co-chaperones Tim14 (Pam18), Tim16 (Pam16), and Mge1 (Craig, 2018; Mokranjac, 2020), to the Tim17–Tim23 core. In the presence of a hydrophobic sorting signal downstream of the presequence, translocation into the matrix is

---

[1]Biocenter-Department of Cell Biology, LMU Munich, Munich, Germany   [2]Department of Parasitology, Faculty of Science, Charles University, BIOCEV, Vestec, Czech Republic   [3]Faculty of Life Sciences and Institute for Protein Dynamics, Kyoto Sangyo University, Kyoto, Japan

Correspondence: mokranjac@bio.lmu.de

stalled and the TM segment is inserted laterally into the IM in a reaction supported by Mgr2 (Chacinska et al, 2005; Popov-Celeketić et al, 2008; Ieva et al, 2014).

Though all the essential components of the presequence pathway have likely been already identified, very little is known about the molecular mechanisms of their function. In particular, we have a very poor understanding of how the TOM and TIM23 complexes cooperate in the IMS to translocate proteins across two mitochondrial membranes in a coordinated fashion. Tim50 was identified as the main receptor of the TIM23 complex and was thus implicated in the transfer of presequence-containing precursor proteins between TOM and TIM23 complexes, yet the molecular understanding of this process is still missing (Geissler et al, 2002; Yamamoto et al, 2002; Mokranjac et al, 2003; Tamura et al, 2009; Shiota et al, 2011; Waegemann et al, 2015; Araiso et al, 2019). Tim50 is synthesized with a cleavable N-terminal presequence and contains a short, matrix-exposed segment followed by a TM segment and a large segment in the IMS (Fig 1A). Though the matrix-exposed segment and the TM of Tim50 were recently implicated in the coordination of the presequence recognition and motor coupling under conditions when full mitochondrial activity is required (Schendzielorz et al, 2017; Caumont-Sarcos et al, 2020), it is the IMS-exposed segment of Tim50 that fulfils the essential functions of this protein as it is sufficient to replace the full-length protein (Mokranjac et al, 2009). In intact mitochondria, Tim50 can be cross-linked to precursor proteins arrested at the *trans* site of the TOM complex (Yamamoto et al, 2002; Mokranjac et al, 2003), in a reaction that depends on Tim50's interaction with the IMS-exposed segment of Tim23 (Gevorkyan-Airapetov et al, 2009; Mokranjac et al, 2009; Tamura et al, 2009). Tim50 can also be cross-linked to the IMS-exposed segments of Tom22 and Tom40 (Tamura et al, 2009; Shiota et al, 2011; Waegemann et al, 2015; Araiso et al, 2019). In vitro binding assays with the recombinantly purified IMS segment of Tim50 and presequence peptides recapitulated the receptor function of Tim50 (Marom et al, 2011; Schulz et al, 2011; Lytovchenko et al, 2013; Rahman et al, 2014). Interestingly, an in vitro study mapped the presequence-binding site to the C-terminal part of the IMS segment of Tim50 (Schulz et al, 2011) which led to the proposal that the IMS segment of Tim50 consists of two domains—the previously crystallized core domain (aa 133–365) (Qian et al, 2011) and the C-terminal segment that contains the presequence-binding domain (PBD) (aa 366–476) (Schulz et al, 2011). Despite the fact that the core domain can bind presequences on its own in vitro (Lytovchenko et al, 2013) and that it contains all the so far identified positions which, when mutated, impair Tim50's interaction with Tim23 (Tamura et al, 2009; Qian et al, 2011; Dayan et al, 2019), deletion of only PBD is already lethal for yeast cells (Schulz et al, 2011). The Tim50 construct lacking the PBD cannot be cross-linked to Tom22 (Waegemann et al, 2015) and, furthermore, the PBD was proposed to be required for an efficient complex formation with the IMS segment of Tim23 (Bajaj et al, 2014; Gomkale et al, 2021). Taken together, these results put the functional relevance of the core domain of Tim50 in doubt.

In this study, we show that the deletion of either the core domain or the PBD of Tim50 is lethal for yeast cells. Unexpectedly, we found that the two domains of Tim50, when co-expressed *in trans*, can support the function of the full-length protein. With this strain, named 50split, we were able to analyse, in vivo and in organello, the functions of the two domains of Tim50 in the IMS. Our results reveal that the core domain is the main recruitment point of Tim50 to the TIM23 complex and also contains the main presequence-binding

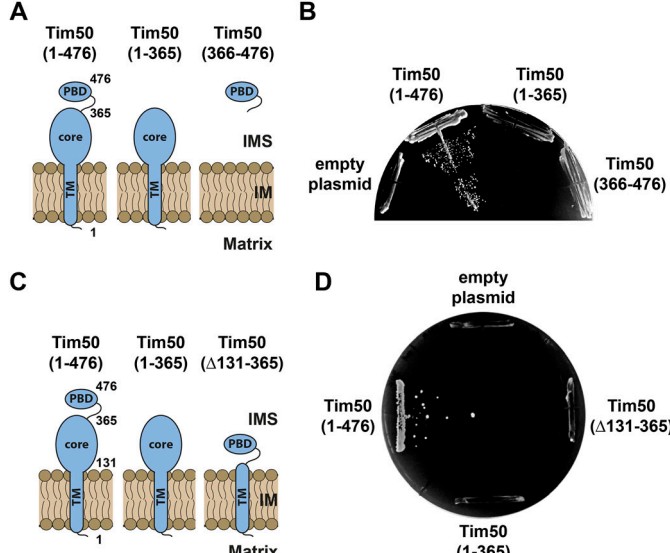

**Figure 1. Both domains of Tim50 in the IMS are essential for cell viability.**
**(A, C)** Schematic representation of the domain structure of Tim50 and the variants analysed. 1 and 476 indicate the first and the last amino acid residues of Tim50 precursor, 131 indicates the start of the IMS segment, and 365 indicates the last residue of the core domain. TM, transmembrane segment; PBD, presequence-binding domain; IMS, intermembrane space; IM, inner membrane. **(B, D)** A Tim50 shuffling strain was transformed with centromeric plasmids encoding indicated variants of Tim50 under control of the endogenous promoter and 3′UTR. The ability of the Tim50 variants to support the function of the full-length protein was analysed on plates containing 5-FOA. Empty plasmid and a plasmid encoding a wild-type copy of Tim50 were used as negative and positive controls, respectively.

site. The PBD, on the other hand, supports the receptor function of Tim50 and plays, directly or indirectly, an important role in cooperation of the TOM and TIM23 complexes. Thus, the two domains of Tim50 both have critical but distinct roles and together coordinate the translocation of proteins across the TOM and TIM23 complexes.

## Results

### Both domains of Tim50 in the IMS are essential for cell viability

To gain molecular insight into the functions of the two domains of Tim50 in the IMS, we first analysed the ability of the individual domains to support the function of the full-length Tim50. To this end, we generated two Tim50 variants (Fig 1A). In the first one, Tim50(1–365), Tim50 was truncated from the C-terminus so that the PBD was deleted. In the second one, Tim50(366–476), only the PBD was present. Because the targeting information of Tim50 is present in its N-terminal segments that are absent in the second variant, the PBD was targeted to the IMS using the first 167 residues of yeast cytochrome $b_2$. We transformed the Tim50 variants into a Tim50 shuffling strain and plated the cells on a medium containing 5-fluoroorotic acid (5-FOA) to remove the *URA* plasmid carrying the wild-type copy of Tim50. Whereas cells transformed with the plasmid encoding the wild-type, full-length Tim50, Tim50(1–476), grew on the 5-FOA medium, neither of the two Tim50 variants gave viable clones, like the empty plasmid which was

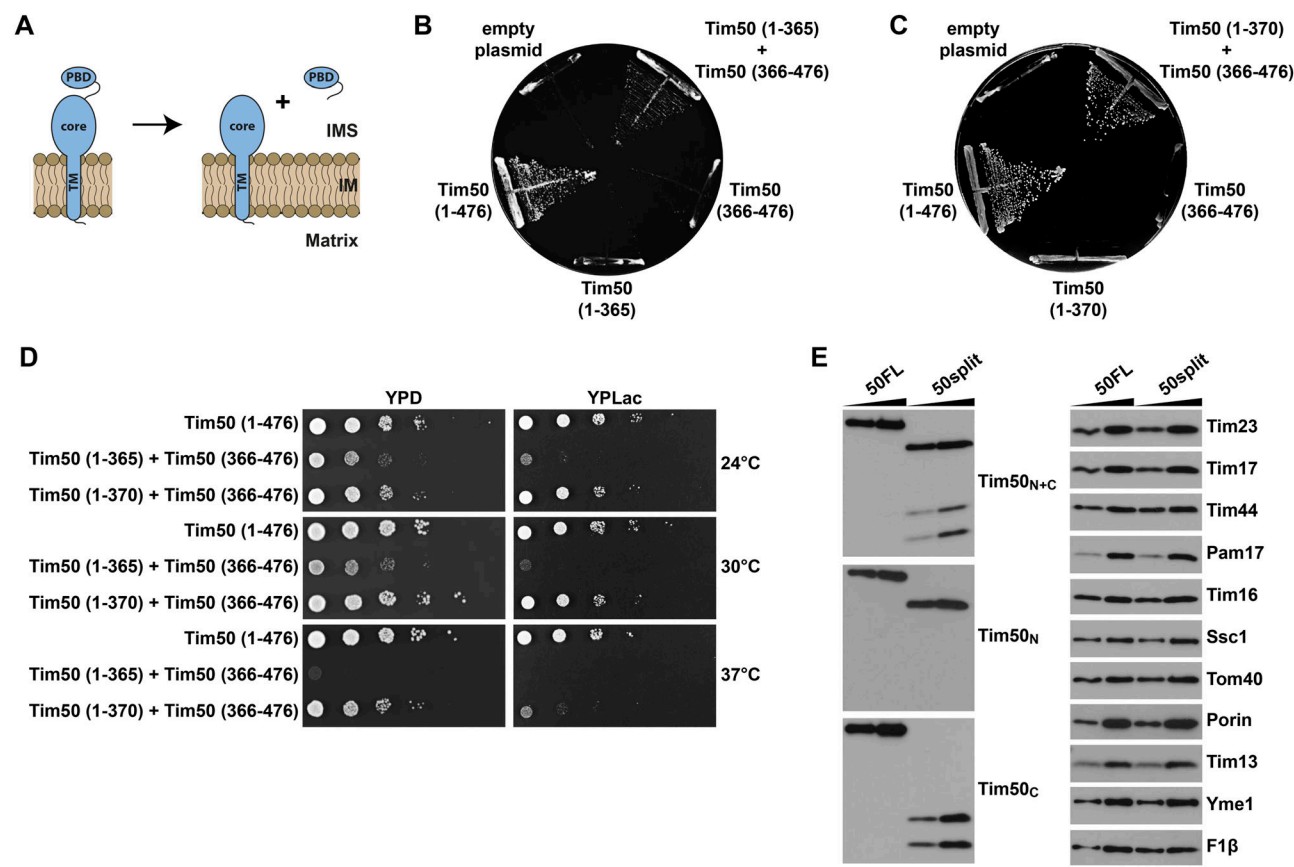

**Figure 2. The function of Tim50 can be rescued by its two domains, core and PBD, expressed *in trans*.**
**(A)** Schematic representation of the Tim50 variants expressed *in trans*. **(B, C)** A Tim50 shuffling strain was transformed with centromeric plasmids carrying the indicated Tim50 variants under control of endogenous promoter and 3'UTR. The ability of the Tim50 variants, alone or upon co-expression, to support the function of the full-length protein was analysed on plates containing 5-FOA at 24°C (B) or at 30°C (C). Empty plasmid and a plasmid encoding a WT copy of Tim50 were used as negative and positive controls, respectively. **(D)** Growth of indicated yeast strains was analysed by 10-fold serial dilution spot assay on plates containing rich medium with glucose (YPD) or lactate (YPLac), as fermentable and non-fermentable carbon sources, respectively. **(E)** Isolated mitochondria (10 and 20 µg) from 50FL and 50split cells were analysed by SDS–PAGE, followed by immunoblotting against depicted mitochondrial proteins. For simplicity reasons, the co-expression strain Tim50(1–370) + Tim50(366–476) was named "50split" and the corresponding wild-type strain expressing the full-length version of Tim50, Tim50(1–476), "50FL."

used as a negative control (Fig 1B). To exclude the possibility that the inability of the PBD to rescue the function of Tim50 was because of the lack of the endogenous Tim50 presequence and its TM segment, we generated the third Tim50 variant, Tim50(Δ131–365), in which only the core domain was deleted, and therefore, the PBD was anchored to the IM with the endogenous TM of Tim50 (Fig 1C). However, also after transformation of this Tim50 variant into the Tim50 shuffling strain and subsequent 5-FOA chase, no viable colonies were obtained (Fig 1D). We conclude that both domains of Tim50 in the IMS, core and PBD, are essential for the viability of yeast cells and that neither of the two domains is, on its own, sufficient to support the function of Tim50.

### The function of Tim50 can be reconstituted in vivo from its two domains expressed *in trans*

We wondered whether the function of Tim50 could be reconstituted in vivo by co-expressing its two domains *in trans* (Fig 2A). For this, we co-transformed the Tim50 variant lacking the PBD, Tim50(1–365), together with the IMS-targeted PBD, Tim50(366–476), into the Tim50 shuffling strain and analysed the ability of yeast cells to grow on a

medium containing 5-FOA. We found that the two domains of Tim50, when co-expressed *in trans*, gave viable colonies on 5-FOA plates incubated at 24°C, whereas neither of the individual Tim50 variants was able to support the function of the full-length Tim50 on its own under the same conditions (Fig 2B). The rescue was dependent on anchoring the core domain to the IM, because co-expression of a soluble, IMS-targeted core domain, Tim50(132–365), together with either a soluble or IM-anchored PBD, Tim50(366–476), and Tim50(Δ164–365), respectively, did not produce any viable yeast cells on the 5-FOA medium (Fig S1A–D). Interestingly, also the cells in which both the core and PBD were anchored to the IM with the endogenous TM of Tim50 were not viable (Fig S1E and F), suggesting that the TM of Tim50 may specifically pack with the TMs of other TIM23 subunits so that the presence of more than one such TM may interfere with the assembly of the complex and/or recruitment of Tim50 to the complex.

We made a serendipitous observation that extending the core domain of Tim50 by just five residues in the Tim50(1–370) variant, upon co-expression with Tim50(366–476), produced viable yeast cells on 5-FOA plates even at 30°C (Fig 2C). Tim50(1–370) on its own, however, still did not yield any viable colonies, just like its shorter

version, Tim50(1–365). The difference in growth between the two Tim50 co-expression strains was even more obvious in serial dilution spot assay (Fig 2D). Whereas, the initial Tim50 co-expression strain, Tim50(1–365) + Tim50(366–476), only grew on a fermentable medium at 24°C and 30°C and even there only poorly, the second Tim50 co-expression strain, Tim50(1–370) + Tim50(366–476), grew like the corresponding wild-type strain at 24°C and 30°C on both fermentable and non-fermentable media (Fig 2D). Even at an elevated temperature (37°C), the second Tim50 co-expression strain still grew almost like the wild-type on the fermentable medium and only showed an obvious growth defect on the non-fermentable medium.

In conclusion, the function of Tim50 can be reconstituted in vivo from its two IMS domains expressed in trans, enabling a dissection of the roles of the two domains of Tim50 in the IMS. For simplicity reasons, we named the strain co-expressing Tim50(1–370) and Tim50(366–476), "50split" and will use this name hereafter, in comparison to "50FL," the corresponding wild-type strain expressing the full-length version of Tim50, Tim50(1–476).

To confirm that the full-length Tim50 is indeed absent in 50split cells and analyse the potential effects of splitting Tim50 on the expression of other mitochondrial proteins, we isolated mitochondria from 50FL and 50split cells and compared their mitochondrial proteins profiles using SDS–PAGE and Western blot. Immunostaining with antibodies raised against the peptides that correspond either to the first 15 amino acid residues of mature Tim50 (Tim50$_N$) or to its last 15 amino acid residues (Tim50$_C$) revealed no full-length Tim50 protein in 50split mitochondria (Fig 2E). This demonstrates that the full-length Tim50 is indeed not necessary for growth of yeast cells. Instead of the full-length Tim50, three faster migrating protein species were detected in mitochondria isolated from 50split cells (Fig 2E). The Tim50$_N$ antibodies specifically detected the matrix-exposed segment of the IM-anchored core domain of the Tim50(1–370) variant, whereas the Tim50$_C$ antibodies specifically recognized two faster migrating protein species that correspond to the IMS-targeted PBD of Tim50 in the Tim50(366–476) variant. The presence of the two protein bands detected for the PBD is likely because of the incomplete processing of the cytochrome $b_2$ sorting signal. The expression levels of the two Tim50 variants in 50split were comparable with the full-length Tim50 in 50FL mitochondria. Splitting of Tim50 also did not affect the expression of other subunits of the TIM23 complex analysed (Tim23, Tim17, Tim44, Pam17, Tim16, Ssc1), nor of any other analysed mitochondrial proteins present in the OM (Tom40, porin), IMS (Tim13), IM (Yme1) or the matrix (F1$\beta$). We conclude that the full-length Tim50 can be split into two segments that are able to restore the functions of Tim50 when co-expressed in trans, and that this does not affect the levels of other mitochondrial proteins.

## Tim50 is recruited to the TIM23 complex mainly through its core domain

It is, in principle, possible that the 50split cells are viable because the two individually expressed segments of Tim50 interact strongly with each other. To analyse this possibility and investigate how Tim50 is recruited to the TIM23 complex in 50split cells, we performed a coimmunoprecipitation experiment. After solubilization of 50FL mitochondria with digitonin, affinity-purified Tim50$_N$ and Tim50$_C$

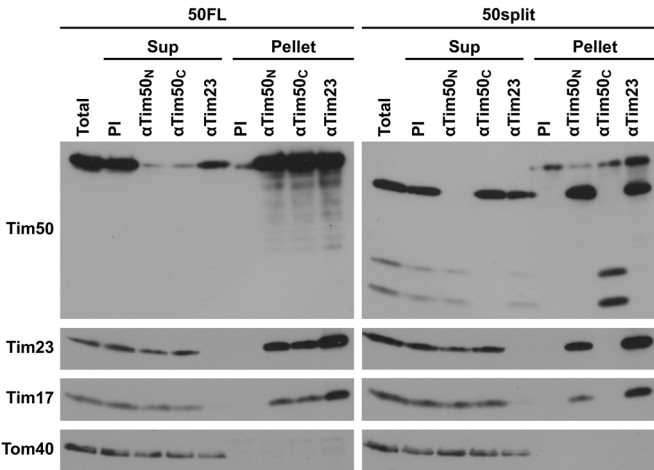

**Figure 3.  Tim50 is recruited to the TIM23 complex mainly through its core domain.**
Isolated mitochondria from 50FL and 50split cells were solubilized with digitonin-containing buffer and subjected to immunoprecipitation with affinity-purified antibodies against Tim50$_N$, Tim50$_C$, and Tim23 prebound to Protein A Sepharose beads. Antibodies from pre-immune serum (PI) were used as a negative control. After washing, specifically bound proteins were eluted with Laemmli buffer. Total (20%), supernatant (Sup, 20%), and bound (Pellet, 100%) fractions were analysed by SDS–PAGE and immunoblotting with indicated antibodies. (*) indicates the heavy chains of the IgGs.

antibodies essentially depleted the full-length Tim50 from the lysate and coprecipitated both Tim23 and Tim17 to a similar extent (Fig 3). On the other hand, affinity-purified antibodies against Tim23 depleted both Tim23 and Tim17 from the lysate and coprecipitated a fraction of Tim50. When digitonin-solubilized 50split mitochondria were analysed, affinity-purified Tim50$_N$ antibodies depleted the fragment of Tim50 which contains the core domain but did not coprecipitate any PBD. Similarly, Tim50$_C$ antibodies depleted the PBD of Tim50 from the 50split mitochondrial lysate but did not coprecipitate any core domain. Thus, the two domains of Tim50 do not interact strongly with each other, at least not even under the mild solubilization conditions used here. When coprecipitation of Tim23 and Tim17 from 50split mitochondrial lysate was analysed, the two proteins were found in the bound fraction only when Tim50$_N$ antibodies were used for immunoprecipitation and not when Tim50$_C$ antibodies were used. Similarly, antibodies to Tim23 only coprecipitated the fragment of Tim50 which contains the core domain and not the PBD. The interaction between Tim17 and Tim23 was not affected by splitting of Tim50. We also did not detect any Tom40 in the immunoprecipitated fractions with either Tim50$_N$ or Tim50$_C$ antibodies, neither in 50split nor in 50FL mitochondria, suggesting that neither of the two Tim50 fragments interacts stably with the TOM complex. Together, these results demonstrate that the two domains of Tim50 do not stably interact with each other and that Tim50 is recruited to the TIM23 complex mainly through its core domain.

## Protein import via the TIM23 complex and binding of Tim50 to precursors are impaired in 50split cells

Considering the essential role of Tim50 during translocation of precursor proteins into the mitochondria, we analysed how splitting of Tim50 affects protein import. Several artificial and endogenous

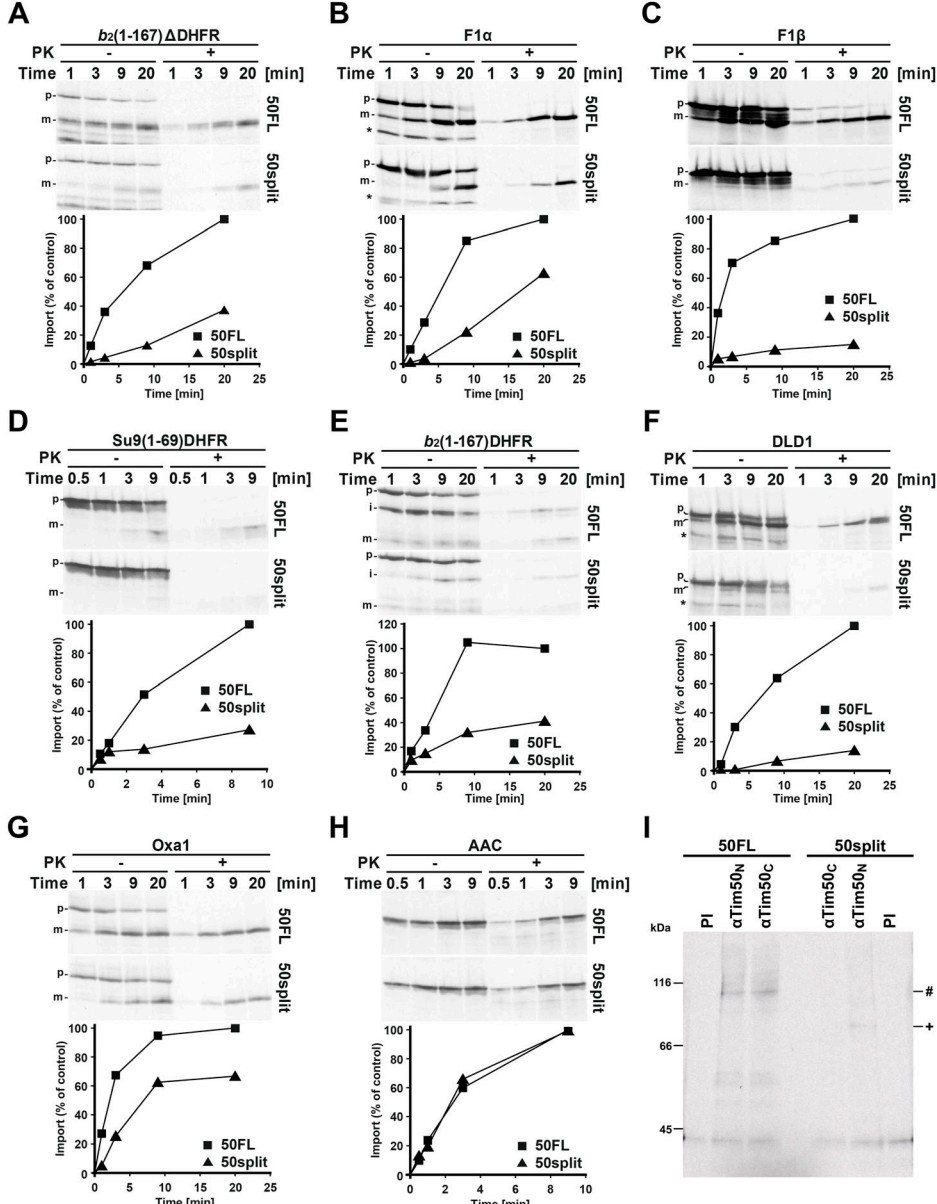

**Figure 4. Protein import via the TIM23 complex and binding of Tim50 to precursors is impaired in 50split cells.**

**(A, B, C, D, E, F, G, H)** $^{35}$S-labelled mitochondrial precursor proteins were imported into the mitochondria isolated from 50FL and 50split cells. After indicated time periods, aliquots were taken, import was stopped, and Proteinase K (PK) was added, where indicated. Mitochondria were reisolated and analysed by SDS–PAGE and autoradiography (upper panels). Quantifications of PK-protected mature forms of imported proteins are shown in the lower panels. The amount of the PK-protected mature form of imported proteins in the longest time point in 50FL mitochondria was set to 100%. Precursor (p), intermediate (i), and mature (m) forms of imported proteins. (*) indicates translation products synthesized from an internal methionine. **(I)** $^{35}$S-labelled Oxa1 precursor was imported into isolated 50FL and 50split mitochondria in the absence of membrane potential. Samples were subjected to cross-linking with 1,5-difluor-2,4-dinitrobenzol (DFDNB). After quenching of excess cross-linker, mitochondria were reisolated and solubilized in SDS-containing buffer to dissociate all noncovalent interactions. Samples were diluted in Triton X-100-containing buffer and subjected to immunoprecipitation with affinity-purified antibodies against N- (Tim50$_N$) and C-terminal peptides (Tim50$_C$) of Tim50 prebound to Protein A Sepharose. Antibodies from pre-immune serum (PI) were used as a negative control. The immunoprecipitates were analysed by SDS–PAGE and autoradiography. (#) and (+) indicate the immunoprecipitated cross-linking adducts of the Oxa1 precursor with full-length Tim50 and the core domain of Tim50, respectively.

precursor proteins were $^{35}$S-labelled in vitro and incubated with isolated 50split and 50FL mitochondria (Fig 4A–H). Protein import via the TIM23 complex into 50split mitochondria was strongly impaired compared with 50FL mitochondria, irrespective of whether translocation into the matrix ($b_2$[1–167]ΔDHFR, Su9[1–69]DHFR, F1α and F1β) or lateral insertion ($b_2$[1–167]DHFR and DLD1) was analysed. Translocation of the TIM23 complex-dependent, presequence-containing precursor protein Oxa1, which contains multiple TM segments, was similarly impaired. On the other hand, the ADP/ATP carrier (AAC) that does not use the TIM23 complex for its translocation into mitochondria, was imported with the same efficiency into 50split as in 50FL mitochondria. Taken together, these results demonstrate that splitting of Tim50 impairs import of precursor proteins via the presequence pathway and that the observed import defects are not because of a general dysfunction of mitochondria but rather are the consequence of impaired Tim50 function.

Tim50 is the first component of the TIM23 complex that interacts with the incoming presequences, as soon as they appear at the outlet of the TOM complex. To analyse how splitting of Tim50 affects the receptor function of the protein and which of the two domains in the IMS recognizes the presequences in organello, we imported $^{35}$S-labelled Oxa1 precursor into 50FL and 50split mitochondria in the absence of membrane potential and incubated the samples with the cross-linking reagent DFDNB. After quenching the excess of cross-linker and solubilization of mitochondria with SDS to dissociate all noncovalent interactions, mitochondrial lysates were subjected to immunoprecipitation with affinity-purified Tim50$_N$ and Tim50$_C$ antibodies and with antibodies from a pre-immune serum, as a negative control. In 50FL mitochondria, cross-linking adducts between Oxa1 and the full-length Tim50 were efficiently precipitated with both Tim50$_N$ and Tim50$_C$ antibodies, demonstrating

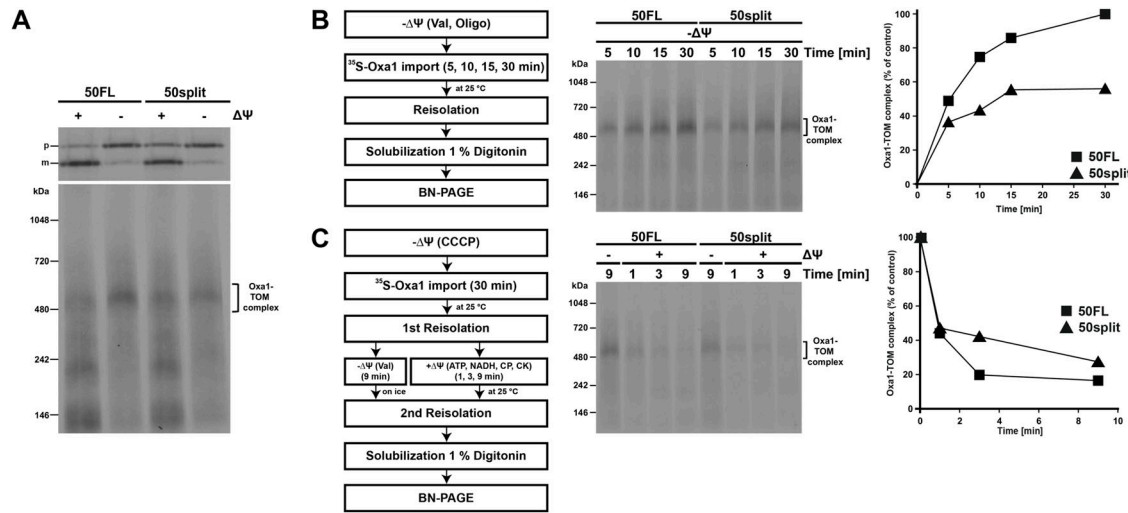

**Figure 5. Association of precursor proteins with the TOM complex is already affected in 50split cells.**
**(A)** $^{35}$S-labelled Oxa1 precursor was imported into 50FL and 50split mitochondria at 25°C in the presence or in the absence of membrane potential (ΔΨ), as indicated. Mitochondria were reisolated, solubilized in digitonin-containing buffer, and samples were analysed by SDS–PAGE (upper panel) and BN-PAGE (lower panel) followed by autoradiography. p, precursor and m, mature forms of Oxa1. **(B)** $^{35}$S-labelled Oxa1 precursor was imported into 50FL and 50split mitochondria in the absence of ΔΨ. Samples were taken at indicated time points, mitochondria were reisolated, solubilized with digitonin, and samples were analysed on BN-PAGE and autoradiography (middle panel). Right panel, quantification of the Oxa1–TOM complex intermediate. The amount of the intermediate at the latest time point in 50FL was set to 100%. **(C)** $^{35}$S-labelled Oxa1 precursor was incubated with 50FL and 50split mitochondria in the absence of ΔΨ. After reisolation, the mitochondria were either kept with dissipated ΔΨ or were energized to chase Oxa1 into the mitochondria. At indicated time points, mitochondria were reisolated again, solubilized in digitonin-containing buffer, and analysed as in panel (B). The amounts of Oxa1–TOM complex intermediates in the samples kept without membrane potential were set to 100%.

that both antibodies are, in principle, capable of detecting the cross-links between Tim50 and this precursor. In 50split mitochondria, however, the cross-link was only visible when immunoprecipitation was done with Tim50$_N$ and not with Tim50$_C$ antibodies (Fig 4I). This demonstrates that the Oxa1 precursor is, in intact mitochondria, primarily recognized by the core domain of Tim50. It should be noted that the intensity of the cross-linking adduct precipitated with Tim50$_N$ antibodies in 50split mitochondria was decreased compared with the one in 50FL mitochondria, indicating that the receptor function of Tim50 is compromised in 50split mitochondria. Essentially the same result was obtained when another precursor protein, $b_2$(1–167)ΔDHFR$_{K5}$, was used, confirming that the core domain of Tim50 is the major interaction point for presequences in intact mitochondria (Fig S2). We conclude that the core domain of Tim50 contains the primary binding site for presequences in intact mitochondria and that the PBD contributes to the receptor function of Tim50.

## Association of precursor proteins with the TOM complex is already affected in 50split cells

As the interaction of Tim50 with precursor proteins is impaired in 50split mitochondria at the stage when the major part of the precursor protein is still in the TOM complex, we analysed whether association of precursor proteins with the TOM complex is also affected in 50split mitochondria. To this end, we imported $^{35}$S-labelled Oxa1 precursor in 50FL and 50split mitochondria in the presence or in the absence of membrane potential, reisolated mitochondria, solubilized them in digitonin-containing buffer, and analysed the samples by SDS- and BN-PAGE. When the samples were analysed by SDS–PAGE, Oxa1 precursor was processed to its mature form in both 50FL and 50split mitochondria in the presence

of membrane potential; however, this processing was more efficient in 50FL mitochondria (Fig 5A, upper panel), in agreement with the protein import defect in 50split mitochondria observed above. In the absence of membrane potential, Oxa1 accumulated in the precursor form in both 50FL and 50split mitochondria. This precursor form of Oxa1 was previously shown to accumulate as an intermediate in the TOM complex that can be resolved by BN-PAGE (Frazier et al, 2003; Chacinska et al, 2005). The Oxa1–TOM complex intermediate was formed less efficiently in 50split compared with 50FL mitochondria (Fig 5A, lower panel). When we analysed the kinetics of Oxa1–TOM complex intermediate formation, we observed that the intermediate was formed more slowly in 50split mitochondria so that after 30 min, it reached only about 50% of the Oxa1–TOM complex intermediate formed in 50FL mitochondria (Fig 5B). Not only was this intermediate formed more slowly in 50split mitochondria but, upon re-establishment of the membrane potential, the Oxa1 precursor was also chased from this stage more slowly in 50split mitochondria compared with 50FL, though the latter effect was less pronounced (Fig 5C).

Taken together, these results indicate that splitting of Tim50 impairs both binding of precursor proteins to the TOM complex and their subsequent transfer to the TIM23 complex. Thus, splitting of Tim50 affects the function of TOM complex in preprotein translocation.

## 50split cells show strong negative genetic interactions with TOM *trans* site mutants

Because the biochemical experiments shown above demonstrated that splitting of Tim50 impairs interactions of precursor proteins with the TOM complex, we investigated whether we could obtain genetic evidence to further support this finding. To explore the

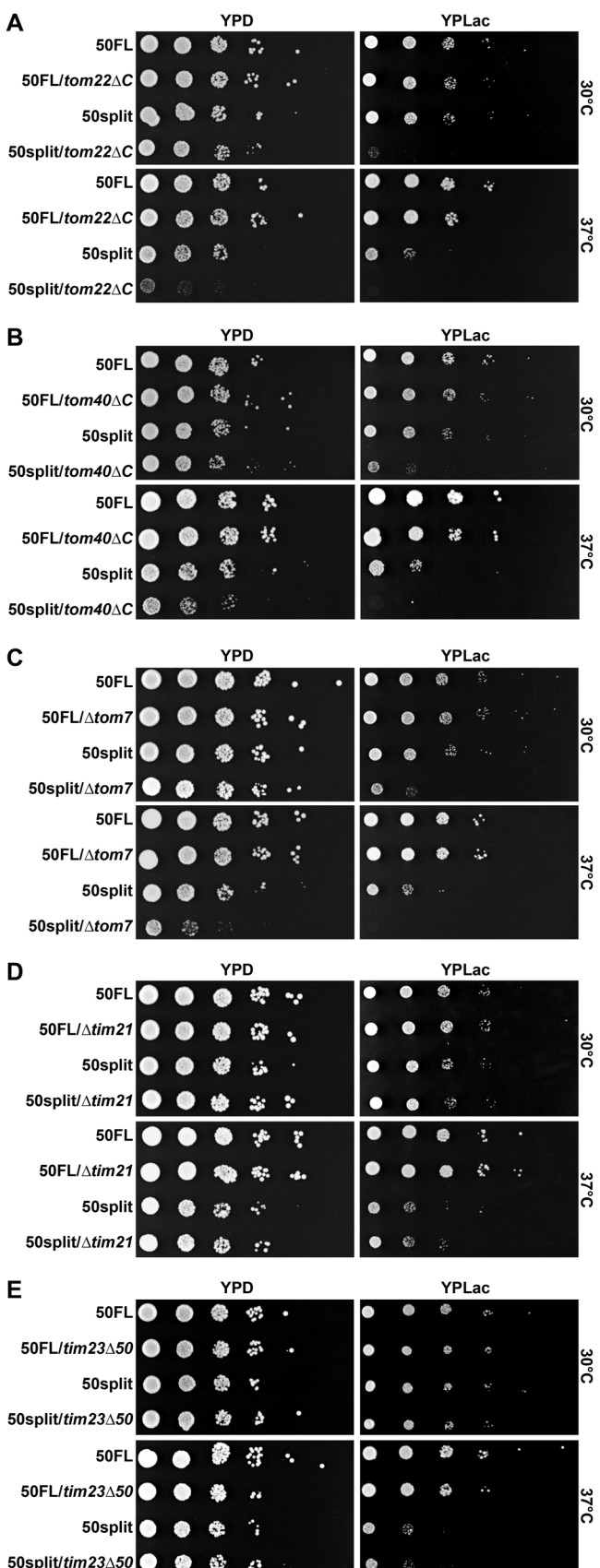

influence of Tim50 splitting on the coordination between the TOM and TIM23 complexes on the genetic level, we generated TOM *trans* site mutants in the background of 50FL and 50split cells. Removal of the IMS-exposed segments of Tom22 (*tom22ΔC*) and Tom40 (*tom40ΔC*) and the deletion of Tom7 (*Δtom7*) in the background of 50FL cells did not visibly impair cell growth on either fermentable or non-fermentable media at any of the temperatures tested (Fig 6A–C). However, combining the same mutations with 50split resulted in cells that grew very poorly on a fermentable medium at 37° C and that were essentially dead on non-fermentable media. Deletion of Tim21 (*Δtim21*), and the Tim23 mutant lacking the N-terminal 50 residues (*tim23Δ50*) did not exacerbate the growth of either 50FL or 50split cells on any of the media or temperatures analysed (Fig 6D and E). In conclusion, both biochemical and genetic evidence suggest that the coordinated translocation of precursor proteins by the TOM and TIM23 complexes is impaired in 50split cells.

## Separation of the two domains of Tim50 in the IMS affects interaction of Tim50 with Tom22

Though the 50split strain is surprisingly viable, all the results shown so far indicate that splitting of Tim50 into core and PBD impairs the function of the protein. We wondered whether it is possible to recapitulate these findings by artificially separating the core and PBD of Tim50 within the same polypeptide chain. To this end, we cloned Tim50 variants in which the endogenous sequence between the core and PBD was replaced with either 15 amino acid residues long flexible or rigid linkers (Kümmel et al, 2011; Chen et al, 2013; Patel et al, 2022) (Fig 7A). Upon transformation into the Tim50 shuffling strain and subsequent 5-FOA chase, yeast cells expressing Tim50 variants containing either of the two different flexible linkers, Tim50flex1 and Tim50flex2, were growing comparable with the cells containing wild-type Tim50 on the fermentable medium and were only slightly impaired in growth on non-fermentable medium. In contrast, cells expressing Tim50 variants containing any of the three rigid linkers, Tim50rig1, Tim50rig2, and Tim50rig3, had severe growth defects on fermentable medium and were virtually dead on the non-fermentable one (Fig 7B). The expression levels of all Tim50 linker mutants were indistinguishable from wild-type (Fig 7C), demonstrating that the observed growth defects were not caused by a general destabilization of the protein. Thus, artificially separating the two domains of Tim50 in the IMS impairs growth of yeast cells even more than splitting them into two polypeptides. Because an impaired TOM–TIM23 cooperation appears to be the main consequence of the impaired Tim50 function in 50split cells, we analysed whether separation of the two Tim50 domains in the IMS by an artificial rigid linker impairs interaction of Tim50 with the TOM complex. For this, we generated Tim50 linker variants in the background of a C-terminally His-tagged Tom22, isolated mitochondria and performed chemical cross-linking with di-thiobis(succinimidyl propionate) (DSP), followed

**Figure 6. 50split cells show strong negative genetic interactions with TOM *trans* site mutants.**
**(A, B, C, D, E)** Growth of the *tom22ΔC, tom40ΔC, Δtom7, Δtim21,* and *tim23Δ50* cells, in the background of either 50FL or 50split, was analysed by 10-fold serial dilution spot assay on plates containing a rich medium with glucose (YPD) or lactate (YPLac), as fermentable and non-fermentable carbon sources, respectively. Plates were incubated at the indicated temperatures.

by Ni-NTA pulldown of His-tagged Tom22 under denaturing conditions so that only cross-linked and therefore covalently bound proteins would be co-isolated along with Tom22. Wild-type Tim50 was efficiently cross-linked to Tom22 and eluted together with His-tagged Tom22 (Fig 7D). Similarly, the Tim50 flexible linker variant was cross-linked to Tom22 and was recovered to a comparable extent like the wild-type Tim50 in the bound fraction. In contrast, the cross-links between the Tim50 rigid linker variant and Tom22 were strongly decreased. Taken together, we conclude that an artificial separation of the two domains of Tim50 in the IMS, either by splitting Tim50 into two polypeptide chains or by a rigid linker, impairs the growth of yeast cells, most likely through impairment of the interaction of Tim50 with the TOM complex.

## Discussion

One of the major unresolved questions of protein import into the mitochondria is how translocation across two mitochondrial membranes is coordinated.

We show here that the two domains of Tim50 in the IMS both have essential roles in translocation of proteins across two mitochondrial membranes. Deletion of either the core domain or PBD of Tim50 is lethal for yeast cells. Based on the unexpected observation that the function of Tim50 can be reconstituted from its two domains expressed *in trans*, we were able to dissect the roles of the individual domains. Our results show that the core domain is the main re-cruitment point of Tim50 to the TIM23 complex, in agreement with the previous findings (Tamura et al, 2009; Qian et al, 2011; Schulz et al, 2011; Dayan et al, 2019). The PBD may, directly or indirectly, support this interaction which could explain the cross-links observed be-tween PBD and Tim23 (Bajaj et al, 2014; Gomkale et al, 2021). Our data also show that, in intact mitochondria, the main binding site for the incoming precursor proteins is present within the core domain of Tim50. However, the receptor function of Tim50 is impaired in 50split cells, suggesting a supporting role for PBD (Schulz et al, 2011; Lytovchenko et al, 2013; Rahman et al, 2014). It is possible that the PBD and the core domain each contribute part of the presequence-binding site or that the presequences are initially recognized by the PBD and then transferred to the core domain. It is, however, also possible that the PBD helps the core to adopt the conformation conducive to recognition of presequences. An artificial separation of the two domains of Tim50 impairs the function of the protein, suggesting that the two domains in IMS need to interact, dynamically or not, with each other. Previous NMR data indeed suggested an interaction between core and PBD of Tim50 (Rahman et al, 2014).

Both the genetic and biochemical pieces of evidence presented here suggest that the coordination of the TOM and TIM23 complexes is particularly sensitive to the separation of the two domains of Tim50 in the IMS. The 50split strain shows very strong negative genetic inter-actions with all segments of the TOM complex exposed to the IMS that were previously implicated in the formation of the *trans* binding site. The negative genetic interactions revealed here of 50split are far stronger than any other TOM–TIM23 contact previously analysed (Waegemann et al, 2015). Furthermore, separation of the core and PBD of Tim50 impairs accumulation of presequence-containing proteins in the TOM complex and their subsequent transfer to the TIM23 complex

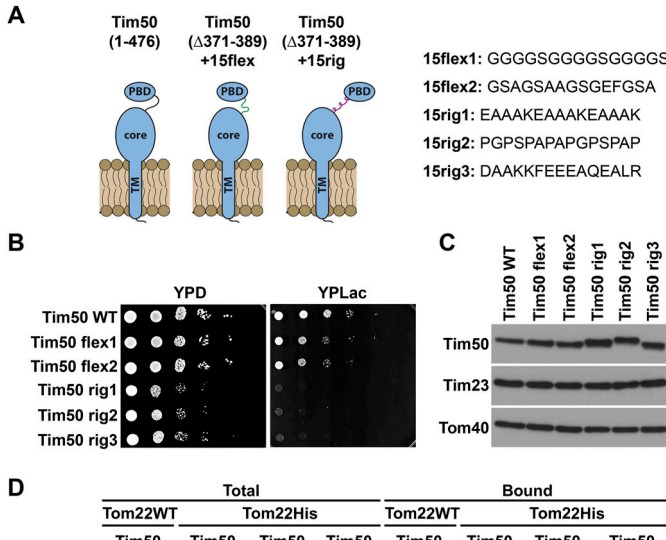

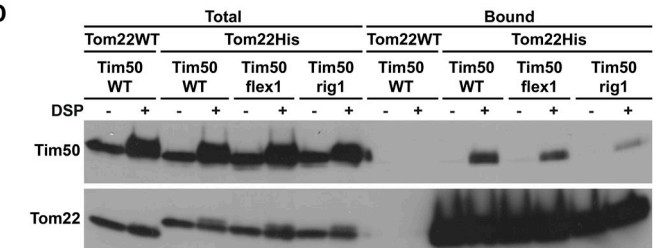

**Figure 7. Separation of the two domains of Tim50 in the IMS impairs growth of yeast cells and affects interaction of Tim50 with Tom22.**
**(A)** Schematic representation (left panel) and amino acid sequences (right panel) of Tim50 linker mutants. **(B)** 10-fold serial dilutions of Tim50 WT and the Tim50 linker mutants were spotted on rich medium-containing glucose (YPD) or lactate (YPLac), as fermentable and non-fermentable carbon sources, respectively. Plates were incubated at 30°C. **(C)** Whole cell extracts were analyzed by SDS–PAGE and immunoblotting. **(D)** Isolated mitochondria, as indicated, were subjected to cross-linking with the amino group-specific and cleavable cross-linker dithiobis(succinimidylpropionate). After quenching of excess cross-linker, mitochondria were reisolated, solubilized in SDS-containing buffer, diluted with Triton X-100 containing buffer, and subsequently incubated with Ni-NTA Agarose beads. After washing, specifically bound proteins were eluted with Laemmli buffer containing 300 mM imidazole and β-mercaptoethanol to cleave the cross-links. Total (5%) and bound fractions (100%) were analysed by SDS–PAGE and immunoblotting with the indicated antibodies.

which are both probably caused by an impaired interaction of Tim50 with the TOM complex. Thus, the essential function of Tim50 is not only to serve as the receptor of the TIM23 complex but also to coordinate translocation across two mitochondrial membranes.

One of the most intriguing but unresolved issues concerning Tim50 is the evolutionary conservation of its domains. It has previously been noted that whereas Tim50 and its core domain are ubiquitously present among eukaryotes, the PBD was so far only identified in fungi (Rahman et al, 2014; Callegari et al, 2020). Con-sidering the ever-increasing number of the available eukaryotic genomes, we performed an extensive bioinformatics analysis of the evolutionary conservation of Tim50. Clear Tim50 orthologues were identified among virtually all sequenced eukaryotic genomes with the exception of the supergroup of Metamonada (Fig S3A and Table S1), which have mitochondria-related organelles with greatly re-duced metabolic capacity (Muñoz-Gómez, 2023). In contrast, PBD was present only among fungi. We cannot completely exclude the possibility that the PBD may be present in non-fungal eukaryotes as

a separate polypeptide, possibly in agreement with the results shown here that, even in yeast, PBD can be separated from the rest of Tim50. However, we used a hidden Markov model-specific PBD to search UniProtKB and EukProt3 databases and these searches returned only fungal species, including microsporidia and other early branching fungi, suggesting that the PBD has evolved early in the evolution of fungi. It is, though, possible that the sequence conservation is so low that the currently available tools are not able to recognize such an ORF as Tim50's PBD, especially considering its very limited length and relatively poor sequence conservation even among fungi. We were also not able to identify any PBD-like extension in any other TOM or TIM23 subunit that could indicate a potential domain-swap event that may have happened in fungi.

PBD of Tim50 is an intriguing example of the differences among the components of the TIM23 complex on the evolutionary scale. The components of the TIM23 complex are extremely well conserved throughout the eukaryotic kingdom and most of them are essential for viability (Mokranjac et al, 2003; Ahting et al, 2009; Demishtein-Zohary & Azem, 2017). Extensions that are sometimes present in yeast, such as the N-terminal extensions present in Tim23 and Tim14, are actively involved in protein import; however, they can be removed without major effects on growth of yeast cells (Donzeau et al, 2000; Chacinska et al, 2003; Mokranjac et al, 2007; Popov-Celeketić et al, 2008; Popov-Čeleketić et al, 2011; Günsel et al, 2020). PBD of Tim50 seems to be an exception—it is present only in fungi, yet, it has an essential role there. How can this be explained? The data presented here show that splitting of the IMS segment of yeast Tim50 into its two domains or their artificial separation through a rigid linker appear to have minor effects on the recruitment of Tim50 to the TIM23 complex but to very strongly influence the ability of Tim50 to coordinate TOM–TIM23 cooperation. It is tempting to speculate that such an elaborate coordination of TOM and TIM23 complexes is particularly important for fast-growing species such as the ones found among fungi. Under optimal growth conditions, yeast cells double their mass, including their mitochondrial content, within 90 min, some *Kluyveromyces* species even faster (within ca. 70 min) (Groeneveld et al, 2009). Therefore, it is not unreasonable to assume that they need to have evolved more efficient mitochondrial protein import systems than the slower growing eukaryotes. It is likely that the coordination of protein translocation across two mitochondrial membranes represents one of the critical, rate-limiting steps that needed to be optimized for an extreme import efficiency. We also found that yeast Tim50 dimerizes through its PBD (Fig S3B and C). This may represent a further adaptation towards increased import efficiency as it would enable the recruitment of two TIM23 complexes to a dimeric TOM complex and thereby also a more efficient transfer of precursor proteins from the outer to the inner mitochondrial membrane. Because Tim23 was also previously shown to dimerize (Bauer et al, 1996; Alder et al, 2008b; Popov-Celeketić et al, 2008; Günsel et al, 2020), it appears that the TIM23 complex has several ways to increase occupancy at the *trans* site of the TOM complex.

Based on all the available data, we propose that the core domain of Tim50 serves both as the main presequence-recognition and -binding element of Tim50 and the main recruitment point to the TIM23 complex, in agreement with its ubiquitous presence among eukaryotes. PBD on the other hand has, directly or indirectly, an important role in TOM–TIM23 coordination, which is apparently essential for fast-growing organisms like fungi but may be less critical for most of the eukaryotes.

# Materials and Methods

**Antibodies.**

| Antibodies | Source | Identifier |
|---|---|---|
| Tim50N | Mokranjac et al (2003) | Tim50N pep (affinity purified) |
| Tim50C | Mokranjac et al (2003) | Tim50C pep (affinity purified) |
| mtHsp70 | Sichting et al (2005) | 347 (affinity purified) |
| Tim23 | Neupert laboratory antibodies | Tim23N pep (affinity purified) |
| Tim17 | Neupert laboratory antibodies | Tim17C pep (affinity purified) |
| Tim44 | Banerjee et al (2015) | 388 (affinity purified) |
| Yme1 | Schreiner et al (2012) | Yme1C pep (affinity purified) |
| Tim16 | Kozany et al (2004) | 335 (affinity purified) |
| Pam17 | Popov-Celeketić et al (2008) | 378 (affinity purified) |
| Porin | Neupert laboratory antibodies | 87118 |
| Tom22 | Neupert laboratory antibodies | Tom22N pep (affinity purified) |
| Tom40 | This study | 547 (affinity purified) |
| F1ß | Neupert laboratory antibodies | 421 |
| Tim13 | Neupert laboratory antibodies | 369 |
| Tim50 | Shiota et al (2011) | N/A |
| FLAG | Sigma-Aldrich | Cat. no.: F3165; RRID:AB 259529 |
| HA | MBL Life Science | Cat. No.: M180-3; RRID:AB 10951811 |

**Chemicals, peptides, and recombinant proteins.**

| Chemical | Source |
|---|---|
| 1,5-difluoro-2,4-dinitrobenzene (DFDNB) | Thermo Fisher Scientific |
| $^{35}$S-Methionine | Perkin Elmer |
| 5-Fluoroorotic Acid (5-FOA) | US Biological Life Sciences |
| Adenosin-5'-triphosphat (ATP) | Roche Diagnostics |
| Anti-HA magnetic beads | MBL Life Science |
| Carbonylcyanid-m-chlorphenylhydrazon (CCCP) | Sigma-Aldrich |
| Creatine kinase (CK) | Roche Diagnostics |
| Creatine phosphate (CP) | Sigma-Aldrich |
| Digitonin | Calbiochem |
| DMSO | Sigma-Aldrich |
| DSP | Thermo Fisher Scientific |
| DTT | Carl Roth |
| IPTG | Merck |
| NADH | GERBU |
| Ni-NTA-Agarose | QIAGEN |
| Oligomycin | Sigma-Aldrich |
| Protease Inhibitor Cocktail | Sigma-Aldrich |
| Protein A Sepharose CL-4B | Cytiva |
| Proteinase K (PK) | Roche Diagnostics |
| TEV protease | Promega |
| Triton-X100 | Sigma-Aldrich |
| Valinomycin | Sigma-Aldrich |

## Critical commercial assays

TNT- Transcription-Translation kit (Promega).
4–16% Native PAGE Bis-Tris Gel (Life Technologies).

**Yeast strains.**

| | |
|---|---|
| YPH499 Δ*tim50::HIS3* + pVT-102U-Tim50(1–476) | Mokranjac et al (2009) |
| YPH499 Δ*tim50::HIS3* + pRS314-prom-Tim50(1–476)-flank | This study |
| YPH499 Δ*tim50::HIS3* + pRS314-prom-Tim50(1–365)-flank + pRS315-prom-$b_2$(1–167)-Tim50(366–476)-flank | This study |
| YPH499 Δ*tim50::HIS3* + pRS314-prom-Tim50(1–370)-flank + pRS315-prom-$b_2$(1–167)-Tim50(366–476)-flank | This study |
| YPH499 Δ*tim50::HIS3* + pRS314-prom-Tim50(Δ371–389)+15flex1-flank | This study |
| YPH499 Δ*tim50::HIS3* + pRS314-prom-Tim50(Δ371–389)+15flex2-flank | This study |
| YPH499 Δ*tim50::HIS3* + pRS314-prom-Tim50(Δ371–389)+15rig1-flank | This study |
| YPH499 Δ*tim50::HIS3* + pRS314-prom-Tim50(Δ371–389)+15rig2-flank | This study |
| YPH499 Δ*tim50::HIS3* + pRS314-prom-Tim50(Δ371–389)+15rig3-flank | This study |
| YPH499 Δ*tim50::HIS3* + pRS314-prom-Tim50(1–476)-flank, Δ*tom22::KAN* + pVT-102U-Tom22(1–152) | This study |
| YPH499 Δ*tim50::HIS3* + pRS314-prom-Tim50(1–476)-flank, Δ*tom22::KAN* + pRS317-prom-Tom22(1–152)-flank | This study |
| YPH499 Δ*tim50::HIS3* + pRS314-prom-Tim50(1–476)-flank, Δ*tom22::KAN* + pRS317-prom-Tom22ΔC(1–119)-flank | This study |
| YPH499 Δ*tim50::HIS3* + pRS314-prom-Tim50(1–370)-flank + pRS315-prom-$b_2$(1–167)-Tim50(366–476)-flank, Δ*tom22::KAN* + pVT-102U-Tom22(1–152) | This study |
| YPH499 Δ*tim50::HIS3* + pRS314-prom-Tim50(1–370)-flank + pRS315-prom-$b_2$(1–167)-Tim50(366–476)-flank, Δ*tom22::KAN* + pRS317-prom-Tom22(1–152)-flank | This study |

**Yeast strains. Continued**

| | |
|---|---|
| YPH499 $\Delta tim50::HIS3$ + pRS314-prom-Tim50(1–370)-flank + pRS315-prom-$b_2$(1–167)-Tim50(366–476)-flank, $\Delta tom22::KAN$ + pRS317-prom-Tom22$\Delta$C(1–119)-flank | This study |
| YPH499 $\Delta tim50::HIS3$ + pRS314-prom-Tim50(1–476)-flank, $\Delta tom40::KAN$ + pVT-102U-Tom40(1–387) | This study |
| YPH499 $\Delta tim50::HIS3$ + pRS314-prom-Tim50(1–476)-flank, $\Delta tom40::KAN$ + pRS317-prom-Tom40(1–387)-flank | This study |
| YPH499 $\Delta tim50::HIS3$ + pRS314-prom-Tim50(1–476)-flank, $\Delta tom40::KAN$ + pRS317-prom-Tom40$\Delta$C(1–363)-flank | This study |
| YPH499 $\Delta tim50::HIS3$ + pRS314-prom-Tim50(1–370)-flank + pRS315-prom-$b_2$(1–167)-Tim50(366–476)-flank, $\Delta tom40::KAN$ + pVT-102U-Tom40(1–387) | This study |
| YPH499 $\Delta tim50::HIS3$ + pRS314-prom-Tim50(1–370)-flank + pRS315-prom-$b_2$(1–167)-Tim50(366–476)-flank, $\Delta tom40::KAN$ + pRS317-prom-Tom40(1–387)-flank | This study |
| YPH499 $\Delta tim50::HIS3$ + pRS314-prom-Tim50(1–370)-flank + pRS315-prom-$b_2$(1–167)-Tim50(366–476)-flank, $\Delta tom40::KAN$ + pRS317-prom-Tom40$\Delta$C(1–363)-flank | This study |
| YPH499 $\Delta tim50::HIS3$ + pRS314-prom-Tim50(1–476)-flank, $\Delta tom7::KAN$ | This study |
| YPH499 $\Delta tim50::HIS3$ + pRS314-prom-Tim50(1–370)-flank + pRS315-prom-$b_2$(1–167)-Tim50(366–476)-flank, $\Delta tom7::KAN$ | This study |
| YPH499 $\Delta tim50::HIS3$ + pRS314-prom-Tim50(1–476)-flank, $\Delta tim21::KAN$ | This study |
| YPH499 $\Delta tim50::HIS3$ + pRS314-prom-Tim50(1–370)-flank + pRS315-prom-$b_2$(1–167)-Tim50(366–476)-flank, $\Delta tim21::KAN$ | This study |
| YPH499 $\Delta tim50::HIS3$ + pRS314-prom-Tim50(1–476)-flank, $\Delta tim23::KAN$ + pVT-102U-Tim23(1–222) | This study |
| YPH499 $\Delta tim50::HIS3$ + pRS314-prom-Tim50(1–476)-flank, $\Delta tim23::KAN$ + pRS317-prom-Tom23(1–222)-flank | This study |
| YPH499 $\Delta tim50::HIS3$ + pRS314-prom-Tim50(1–476)-flank, $\Delta tim23::KAN$ + pRS317-prom-Tom23(51–222)-flank | This study |
| YPH499 $\Delta tim50::HIS3$ + pRS314-prom-Tim50Hi(1–370)-flank + pRS315-prom-$b_2$(1–167)-Tim50(366–476)-flank, $\Delta tim23::KAN$ + pVT-102U-Tim23(1–222) | This study |
| YPH499 $\Delta tim50::HIS3$ + pRS314-prom-Tim50Hi(1–370)-flank + pRS315-prom-$b_2$(1–167)-Tim50(366–476)-flank, $\Delta tim23::KAN$ + pRS317-prom-Tim23(1–222)-flank | This study |
| YPH499 $\Delta tim50::HIS3$ + pRS314-prom-Tim50(1–370)-flank + pRS315-prom-$b_2$(1–167)-Tim50(366–476)-flank, $\Delta tim23::KAN$ + pRS317-prom-Tim23(51–222)-flank | This study |
| YPH499 $\Delta tom22::KAN$ + pRS314-prom-Tom22(1–152)-flank | Waegemann et al (2015) |
| YPH499 $\Delta tom22::KAN$ + pRS314-prom-Tom22(1–152)-His$_6$-flank, $\Delta tim50::HIS3$ + pVT-102U-Tim50(1–476) | This study |
| YPH499 $\Delta tom22::KAN$ + pRS314-prom-Tom22(1–152)-His$_6$-flank, $\Delta tim50::HIS3$ + pRS315-prom-Tim50(1–476)-flank | This study |
| YPH499 $\Delta tom22::KAN$ + pRS314-prom-Tom22(1–152)-His$_6$-flank, $\Delta tim50::HIS3$ + pRS315-prom-Tim50($\Delta$371–389)+15flex1-flank | This study |
| YPH499 $\Delta tom22::KAN$ + pRS314-prom-Tom22(1–152)-His$_6$-flank, $\Delta tim50::HIS3$ + pRS315-prom-Tim50($\Delta$371–389)+15rig1-flank | This study |
| W303-1A $\Delta tim50::CgHIS3$ + pRS316-Tim50(1–476) | Yamamoto et al (2002) |
| W303-1A $\Delta tim50::CgHIS3$ + pRS315-GAL1-Tim50(E415BPA)-FLAG-His$_8$ + p6xtRNA | This study |
| W303-1A $\Delta tim50::CgHIS3$ + pRS315-GAL1-Tim50(E415BPA)-FLAG-His$_8$ + pRS316-prom-Tim50core-TEV-PBD-HA + p6xtRNA | This study |

**Recombinant DNA (plasmids).**

| | |
|---|---|
| pRS314-prom-Tim50(1–476)-flank | This study |
| pRS314-prom-Tim50(1–365)-flank | This study |
| pRS314-prom-Tim50(1–370)-flank | This study |
| pRS315-prom-$b_2$(1–167)-Tim50(366–476)-flank | This study |
| pRS314-prom-Tim50($\Delta$131–365)-flank | This study |
| pRS314-prom-$b_2$(1–167)-Tim50(132–365)-flank | This study |
| pRS315-prom-Tim50($\Delta$164–365)-flank | This study |
| pRS314-prom-Tim50($\Delta$371–389)+15flex1-flank | This study |
| pRS314-prom-Tim50($\Delta$371–389)+15flex2-flank | This study |
| pRS314-prom-Tim50($\Delta$371–389)+15rig1-flank | This study |
| pRS314-prom-Tim50($\Delta$371–389)+15rig2-flank | This study |

## Recombinant DNA (plasmids). Continued

| | |
|---|---|
| pRS314-prom-Tim50(Δ371–389)+15rig3-flank | This study |
| pRS317-prom-Tom40(1–387)-flank | This study |
| pRS317-prom-Tom40ΔC(1–363)-flank | This study |
| pRS317-prom-Tom22(1–152)-flank | This study |
| pRS317-prom-Tom22ΔC(1–119)-flank | This study |
| pRS317-prom-Tim23(1–222)-flank | This study |
| pRS317-prom-Tim23Δ50(51–222)-flank | This study |
| pRS314-prom-Tom22(1–152)-His$_6$-flank | Waegemann et al (2015) |
| pRS315-prom-Tim50(1–476)-flank | This study |
| pRS315-prom-Tim50(Δ371–389)+15flex1-flank | This study |
| pRS315-prom-Tim50(Δ371–389)+15rig1-flank | This study |
| pGEM4-AAC | Banerjee et al (2015) |
| pGEM4-$b_2$(1–167)Δ19DHFR | Banerjee et al (2015) |
| pGEM4-$b_2$(1–167)DHFR | Banerjee et al (2015) |
| pGEM4-DLD1 | Banerjee et al (2015) |
| pGEM4-F1α | This study |
| pGEM4-F1β | Banerjee et al (2015) |
| pGEM4-Oxa1 | Banerjee et al (2015) |
| pGEM4-Su9(1–69)DHFR | Banerjee et al (2015) |
| pRS315-GAL1-Tim50(E415BPA)-FLAG-His$_8$ | This study |
| pRS316-prom-Tim50core-TEV-PBD-HA | This study |
| p6xtRNA | Chin et al (2003) |

## Oligonucleotides (primers).

| | |
|---|---|
| Fw_After50_476flank | TTTTCATGTAAACCCTCTTCTCATG |
| Tim50_365_StoppHind | GGGAAGCTTTTATTTATCCTTCAATTTTTTCACAC |
| Fw_366for50PBD | TTTTACGGAGATCATAAATCTGGTG |
| Revb2_50pRS | GGATCCTTGAAGGGGACCCAATTTTTTC |
| Fw_After50_476flank | TTTTCATGTAAACCCTCTTCTCATG |
| Tim50_365_StoppHind | GGGAAGCTTTTATTTATCCTTCAATTTTTTCACAC |
| Fw_After50_476flank | TTTTCATGTAAACCCTCTTCTCATG |
| Rv_Tim50_370_stopp | TTAATGATCTCCGTAAAATTTATCCTTC |
| Fw_366for50PBD | TTTTACGGAGATCATAAATCTGGTG |
| Rev_50_TM_130 | GTAGATTGCAGTACCTGTCAACGCAGACAACG |
| Fw_366for50PBD | TTTTACGGAGATCATAAATCTGGTG |
| Rev_50_TM_163 | CCTGGCCTTGAATCTTTTATACATAAGTG |
| Fw_Tim50_inflex | GGCGGCGGAGGGTCTGGAGGTGGTGGTAGTGG |
| Rv_Tim50_365+5AS | ATGATCTCCGTAAAATTTATCCTTAAGCTTTTTCACACGATGATC |
| Fw_Tim50_15flex2 | GGTTCTGCCGGCTCCGCTGCCGGTTCAGGAG |
| Rv_Tim50_365+5AS | ATGATCTCCGTAAAATTTATCCTTAAGCTTTTTCACACGATGATC |
| Fw_Tim50_inrig | GAAGCTGCAGCAAAGGAGGCCGCAGCCAAAG |
| Rv_Tim50_365+5AS | ATGATCTCCGTAAAATTTATCCTTAAGCTTTTTCACACGATGATC |

**Oligonucleotides (primers).   Continued**

| | |
|---|---|
| Fw_Tim50_15rig2 | CCAGGTCCATCGCCGGCACCAGCCCCTGGC |
| Rv_Tim50_365+5AS | ATGATCTCCGTAAAATTTATCCTTAAGCTTTTTCACACGATGATC |
| Fw_Tim50_15rig3 | GAAGAAGCTCAAGAAGCTTTGAGAAGCACCAAGTTCCCGCTCGATTTG |
| Rv_Tim50_15rig3 | TTCAAATTTTTTAGCAGCATCATGATCTCCGTAAAATTTATCCTTC |
| Tom40deltaC_f | TGAAAATCTTCCCTTGGCTTTTTTATG |
| Tom40_363_r | ACCAGCAGTTTCAAATTGTAGACC |
| SacTim23p | CCTGAGCTCACTGTGACGTCG |
| Tim23flankEco | CCCGAATTCCAGGTGTTGATCGTTAGCACC |
| Fw476for50dPBD | AAGCTTGCGGCCGCATAATGCTTAAG |
| Rv_Tim50_370_stopp | TTAATGATCTCCGTAAAATTTATCCTTC |
| Sce_tim50_tm_f | GAAGAAAGACATCGATAATGGC |
| BamHI-Tim50f | TAGGATCCATGCTGTCCATTTTAAGAAATTC |
| Tim50FLAGkod_f | TCATGATATCGATTACAAGGATGACGATGACAAGTAACATGTAAACCCTCTTCTCATGTATC |
| Tim50FLAGkod_r | TCTTTATAATCACCGTCATGGTCTTTGTAGTCGAATTCTTTGGATTCAGCAATCTTCTTCTTTTTC |
| SalI-His8-FLAGr | TGATGTCGACCTAATGATGATGGTGATGATGGTGGTGCTTGTCATCGTCATCCTTGTAATCG |
| Tim50amber415 | TAGGAAAAGGAAAAAATTAGAAT |
| Tim50_415r | CTCAATCATCTTCATGAACA |
| EcoRI Tim50 HA r | CATGGAATTCTTACGCGTAGTCTGGAACGTCATATGGGTATTTGGATTCAGCAATCTTCTTCTTTTTC |
| EcoRI_1hato3HAr | CATGGAATTCTTAAGCGTAGTCAGGTACGTCGTAAGGGTAAGCGTA ATCCGGAACGTCGTACGGATACGCGTAGTCTGGAACGTCATATG |
| LoopTEV | GCGGCAGCACCAAGTTCCCGGAAAACCTGTACTTCCAGGGACTCGATTTGATTCATGAAGAAC |

## Software and algorithms (e.g., for quantification)

ImageJ (Fiji).
Excel (Microsoft).
Python.

## Yeast strains, plasmids, and growth conditions

Wild-type haploid yeast strain YPH499 was used for genetic manipulations (Sikorski & Hieter, 1989). A Tim50 shuffling strain in YPH499 background (Mokranjac et al, 2009) was used to generate Tim50 mutant strains. For photocross-linking experiments, wild-type yeast strain W303-1A and a Tim50 shuffling strain in W303-1A were used.

Tim50 variants were cloned into centromeric yeast plasmids pRS314 (*TRP* marker), pRS315 (*LEU* marker) or pRS317 (*LYS* marker) under the control of the endogenous *TIM50* promoter and 3′-untranslated regions (3′UTR). Tim50 variants targeted into the IMS were cloned as fusion proteins, starting with residues 1–167 of yeast cytochrome $b_2$ followed by the indicated Tim50 coding sequence (Mokranjac et al, 2009). The Tim50 variant (pRS315-*GAL1*-Tim50 [E415BPA]-FLAG-His$_8$) for photocross-linking experiments was cloned under the control of a *GAL1* promoter with C-terminal FLAG- and His$_8$-tags. All plasmids were confirmed by sequencing.

For complementation analyses, the Tim50 shuffling strain was transformed with two plasmids simultaneously and the transformants were selected on selective glucose medium lacking the respective markers. Tim50 variants were either co-transformed or,

when transformed individually, empty plasmids were transformed in addition, to have the auxotrophic markers equal among all the strains. The full-length copy of *TIM50* on a plasmid was used as positive control and empty plasmids as negative ones. Cells that lost the wild-type copy of Tim50 on the *URA* plasmid were selected on a medium containing 5-fluoroorotic acid (5-FOA) at 30°C, unless otherwise indicated.

Tom22(1–152), Tom22ΔC(1–119) (Waegemann et al, 2015), Tom40(1–387), Tom40ΔC(1–363), and Tim23Δ50(51–222) (Günsel et al, 2020) constructs were cloned into centromeric yeast plasmid pRS317 (*LYS* marker), including their endogenous promoters and 3′UTRs. These constructs were transformed into the corresponding Tom22, Tom40, and Tim23 shuffling strains, generated in the background of 50FL and 50split cells, and selected on a selective glucose medium lacking the respective markers followed by a 5-FOA chase as described above. The nonessential genes, *TOM7* and *TIM21*, were deleted in the backgrounds of 50FL and 50split cells by replacing them with *KAN* cassettes through homologous recombination. Deletions were confirmed by colony-PCR and Western blotting.

Tim50(1–476), Tim50(Δ371–389)+15flex1 and Tim50(Δ371–389)+15-rig1 were additionally subcloned into centromeric yeast plasmid pRS317 (*LYS* marker). These constructs were transformed into a Tim50 shuffling strain in the background of C-terminally His-tagged Tom22 (YPH499 Δtom22::KAN + pRS314-prom-Tom22[1–152]-His$_6$-flank, Δtim50:: HIS3 + pVT-102U-Tim50[1–476]) and selected on selective glucose medium lacking the respective markers followed by a 5-FOA chase as described above.

Yeast cells were grown in YPD medium at 24°C, unless otherwise indicated. Mitochondria were isolated from cells in the logarithmic growth phase.

For photocross-linking experiments, yeast cells were grown in SC (0.67% yeast nitrogen base without amino acids, 0.5% casamino acids) or S (0.67% yeast nitrogen base without amino acids, 2% lactate and 50 mM $NaH_2PO_4$) media, with appropriate carbon sources (2% galactose [Gal] or 2% sucrose [Suc]), with respective markers and with 1 mM Bpa wherever it is indicated.

For drop dilution spot assays, yeast cells were grown overnight in YPD at 24°C. Overnight cultures were diluted into fresh medium and, when they reached logarithmic growth phase, 10-fold serial dilutions were made and spotted on YPD and YPLac plates, as fermentable and non-fermentable carbon sources, respectively. The plates were incubated at indicated temperatures.

## Coimmunoprecipitation from digitonin-solubilized mitochondria

Mitochondria were solubilized at 1 mg/ml with 1% digitonin in 20 mM TRIS/HCl, 80 mM KCl, 10% glycerol, 2 mM PMSF, pH 8.0 for 15 min at 4°C. Non-solubilized material was removed by ultracentrifugation at 124,500$g$ for 20 min at 4°C and solubilized mitochondria were incubated with indicated antibodies against TIM23 subunits prebound to Protein A Sepharose beads. Antibodies present in the pre-immune serum (PI) served as a negative control. After 45 min incubation at 4°C, nonbound material was collected, beads were washed three times, and bound proteins were eluted with Laemmli buffer. Samples were analysed by SDS–PAGE and immunoblotting.

## Protein import into isolated mitochondria

Precursor proteins were synthesized in the presence of $^{35}$S methionine in a standard or a transcription and translation coupled reticulocyte lysate system (Promega; Mokranjac et al, 2003). Isolated mitochondria were resuspended at 0.5 mg/ml in SI buffer (50 mM HEPES/KOH, 0.6 M sorbitol, 80 mM KCl, 10 mM MgAc$_2$, 2 mM KH$_2$PO$_4$, 2.5 mM MnCl$_2$, 2.5 mM EDTA, pH 7.2), supplemented with 1 mg/ml bovine serum albumin, 2.5 mM ATP, 3.75 mM NADH, 10 mM creatine phosphate, and 100 $\mu$g/ml creatine kinase. Import of $^{35}$S-labelled precursor proteins was performed at 25°C and the reactions were stopped at indicated time points by diluting the samples in ice-cold SH buffer (20 mM HEPES/KOH, 0.6 M sorbitol, pH 7.2) containing 1 $\mu$M valinomycin. One half of the samples were treated with 55 $\mu$g/ml proteinase K (PK) for 15 min on ice. Protease treatment was stopped by adding PMSF to 2 mM. The mitochondria were reisolated by centrifugation at 18,000$g$, 10 min, 4°C, and analysed by SDS–PAGE and autoradiography. Quantifications of the import experiments were done with the ImageJ software. The amount of PK-protected mature form in the longest time point in WT mitochondria was set to 100%. Graphs were created with Python.

## Cross-linking

Cross-linking of precursors to Tim50 in the absence of membrane potential followed by immunoprecipitation was performed, with some modifications, as described before (Mokranjac et al, 2003). To dissipate the membrane potential, mitochondria (0.5 mg/ml) were incubated in SI buffer supplemented with 25 $\mu$M carbonyl cyanide m-chlorophenylhydrazone (CCCP), 8 $\mu$M oligomycin, and 0.5 $\mu$M valinomycin for 15 min at 25°C. $^{35}$S-labelled precursor proteins were then added and the samples were incubated for 15 min at 25°C. The samples were then transferred on ice and the cross-linking was performed for 30 min on ice with 50 $\mu$M homobifunctional, amino group-reactive agent 1,5-difluoro-2,4-dinitrobenzene (DFDNB) freshly dissolved in DMSO. Control samples were treated with DMSO. Cross-linking reaction was stopped by adding glycine, pH 8.8 to 0.1 M. The mitochondria were washed twice with SH buffer and reisolated by centrifugation at 18,000$g$ for 10 min at 4°C. For immunoprecipitation, mitochondria were solubilized at 1 mg/ml with 1% SDS in 50 mM Na-phosphate, 100 mM NaCl, 2 mM PMSF, pH 8.0 for 5 min at 25°C, and diluted in the same buffer containing 0.2% Triton X-100. Non-solubilized material was removed by ultracentrifugation at 124,500$g$ for 20 min at 4°C. Solubilized mitochondria were incubated with affinity purified antibodies against the N-terminal (Tim50$_N$) and C-terminal peptide (Tim50$_C$) of Tim50 prebound to Protein A Sepharose beads. Pre-immune serum (PI) served as a negative control. After 45 min incubation at 4°C, nonbound material was removed, beads were washed three times, and bound proteins were eluted with Laemmli buffer. Samples were analysed by SDS–PAGE followed by autoradiography.

Cross-linking of His-tagged Tom22 to Tim50 followed by Ni-NTA pulldown was performed as described before (Waegemann et al, 2015). Mitochondria (0.5 mg/ml) were incubated in SI buffer containing 2 mM ATP, 2 mM NADH, 10 mM creatine phosphate, and 100 $\mu$g/ml creatine kinase for 3 min at 25°C. Cross-linking was performed with 0.45 mM 3,3'-DSP for 30 min on ice. Cross-linking reaction was stopped by adding 0.1 M glycine, pH 8.8. The mitochondria were washed with SH buffer and reisolated by centrifugation at 18,000$g$ for 10 min at 4°C. For Ni-NTA pulldown of His-tagged Tom22, the mitochondria were solubilized with 1% SDS in 50 mM Na-phosphate, 100 mM NaCl, 10 mM imidazole, 2 mM PMSF, pH 8.0 for 5 min at 25°C and diluted with the same buffer containing 0.2% Triton X-100. Non-solubilized material was removed by ultracentrifugation at 124,500$g$ for 20 min at 4°C. Solubilized mitochondria were subsequently incubated for 45 min at 4°C with Ni-NTA-Agarose beads, pre-equilibrated in the same buffer containing 0.2% Triton X-100. After three washing steps, specifically bound proteins were eluted in Laemmli buffer containing 300 mM imidazole and $\beta$-mercaptoethanol to cleave the cross-links. Samples were analysed by SDS–PAGE and immunoblotting.

## Oxa1 accumulation in the TOM complex

Oxa1 accumulation in the TOM complex was performed, with minor modifications, as described before (Frazier et al, 2003). Mitochondria (0.5 mg/ml) were resuspended in SI buffer containing 1 mg/ml BSA and incubated in the presence of either 20 $\mu$M oligomycin and 1 $\mu$M valinomycin for 10 min at 25°C to deplete the membrane potential or 2.5 mM ATP, 3.75 mM NADH, 10 mM creatine phosphate, 100 $\mu$g/ml creatine kinase for 3 min at 25°C to generate membrane potential. Import of $^{35}$S-labelled Oxa1 precursor was performed at 25°C. The import reaction was stopped after 30 min (or indicated time points) by diluting the samples in ice-cold SH buffer with 1 $\mu$M valinomycin. After reisolation by centrifugation at 18,000$g$,

10 min, 4°C, mitochondria were washed once with SH buffer with 1 $\mu M$ valinomycin. The mitochondria were solubilized at 1 mg/ml with 1% digitonin in 20 mM TRIS, 0.1 mM EDTA, 50 mM NaCl, 10% glycerol, 2 mM PMSF, pH 8.0 for 15 min at 4°C. Non-solubilized material was removed by centrifugation at 13,200$g$ for 15 min at 4°C. Samples were analysed by BN-PAGE and SDS–PAGE followed by autoradiography.

For the chase of Oxa1 precursor into the matrix, membrane potential was dissipated with 50 $\mu M$ carbonyl cyanide m-chlorophenylhydrazone (CCCP) for 10 min at 25°C. After import of $^{35}$S-labelled Oxa1 precursor for 30 min at 25°C, mitochondria were washed with SHK buffer (0.6 M sorbitol, 20 mM HEPES/KOH, 80 mM KCl, pH 7.2) and reisolated by centrifugation for 10 min, 18,000$g$, 4°C. CCCP was washed away by resuspending the mitochondria in SI buffer (50 mM HEPES/KOH, 0.5 M sorbitol, 80 mM KCl, 10 mM MgAc$_2$, 2 mM KH$_2$PO$_4$ and 1 mM MnCl$_2$, 2.5 mM EDTA, pH 7.2, 3% BSA [wt/vol]) containing 5 mM DTT. Membrane potential was re-established by adding 2.5 mM ATP, 3.75 mM NADH, 10 mM creatine phosphate, 100 $\mu g$/ml creatine kinase at 25°C and Oxa1 precursor was chased into the mitochondria. At the indicated time points, samples were taken out and import was stopped by diluting the samples in ice-cold SH buffer with 1 $\mu M$ valinomycin. The control sample was resuspended in SI buffer containing 1 $\mu M$ valinomycin and kept on ice throughout the chase of the Oxa1 precursor. Samples were subsequently handled and analysed as described above.

### BN-PAGE

After solubilization of the mitochondria (as described above), 1 $\mu l$ of 5% Coomassie Brilliant Blue-G was added to 20 $\mu l$ of the solubilized mitochondria. Samples were run according to the manufacturer's instructions on a 4–16% Native PAGE Bis-Tris Gel, blotted onto a PVDF membrane, and analysed by autoradiography.

### Phylogenetic analysis

Tim50 alignment was used for HMMER search (Mistry et al, 2013) against proteomes deposited in EukProt (Richter et al, 2022) and reference proteomes in UniProt (UniProt Consortium, 2023). To make a Tim50 alignment for HMMER search, sequences annotated as Tim50 in UniProt were selected, the incomplete sequences were removed, and the dataset was clustered down using MMseqs2 to remove any possible bias towards overrepresented lineages. To filter out false-positive and incomplete sequences, the sequences identified by HMMER were further analysed for the presence of mitochondrial targeting sequences by TargetP (Emanuelsson et al, 2007) and the transmembrane domain by Tmhmm2 (Krogh et al, 2001).

### In vivo photocross-linking

W303-1A $\Delta tim50::CgHIS3$ + pRS315-*GAL1*-Tim50(E415BPA)-FLAG-His$_8$ + p6xtRNA cells were cultured in SCGal (−Trp, −Ura) medium in the presence or absence of 1 mM BPA. Next, 50 OD$_{600}$ cells were UV-irradiated for 15 min on ice. The cells were suspended in 6 ml of ice-cold TE buffer (10 mM TRIS/HCl, 1 mM EDTA, pH 8.0), followed by addition of 380 $\mu l$ of 5 M sodium hydroxide solution and 420 $\mu l$ of $\beta$-mercaptoethanol. The mixture was then vortexed and incubated

for 10 min on ice. Afterwards, 750 $\mu l$ of 100% trichloroacetic acid was added and the mixture was incubated for another 10 min on ice. The sample was then centrifuged at 17,000$g$ for 10 min and the supernatant was discarded. The pellet was resuspended in 1% SDS (wt/vol) in 50 mM Tris–HCl, pH 8.0, 150 mM NaCl, 1 mM PMSF and heated at 95°C for 5 min. Next, the sample was diluted fourfold in 0.5% Triton X-100 (vol/vol) in 40 mM Tris–HCl, pH 7.5, 200 mM NaCl, and centrifuged at 20,000$g$ for 5 min at 4°C. Ni-NTA agarose was added to the supernatant and incubated on ice for 5 min. After washing the beads with the same buffer, His-tagged BPA containing Tim50 and its cross-linked products were eluted with 1 M imidazole in 50 mM Tris–HCl, pH 8.0, 150 mM NaCl, 100 mM EDTA, 0.5% SDS. Proteins were finally analysed by SDS–PAGE and immunoblotting with antibodies against Tim50.

### In organello photocross-linking and TEV digestion

W303-1A $\Delta tim50::CgHIS3$ + pRS315-*GAL1*-Tim50(E415BPA)-FLAG-His$_8$ + pRS316-prom-Tim50core-TEV-PBD-HA + p6xtRNA cells were cultured in SGalSuc (−Trp, −Ura, -Leu) medium in the presence of 1 mM BPA. Mitochondria were isolated from yeast cells and UV-irradiated for 15 min on ice. The collected mitochondria were solubilized in 1% digitonin in 20 mM TRIS/HCl pH 7.4, 50 mM NaCl, 0.1 mM EDTA, 10% glycerol, 1 mM PMSF, and protease inhibitor cocktail and centrifuged at 20,000$g$ for 5 min at 4°C. Anti-HA magnetic beads were then added to the supernatant and gently rotated for 10 min at 4°C. The beads were washed twice with 20 mM TRIS/HCl pH 7.4, 50 mM NaCl, 0.1 mM EDTA, 10% glycerol, 0.1% digitonin, 1 mM PMSF, and the protease inhibitor cocktail, followed by a single wash with 20 mM TRIS/HCl pH 7.4, 50 mM NaCl, 0.1 mM EDTA, 10% glycerol, 0.1% digitonin. The beads were then mixed with 0.5 $\mu l$ (2.5 U) of TEV protease in 50 $\mu l$ of the same buffer with 1 mM DTT and gently rotated for 2 h at 4°C. Finally, the eluted proteins were analysed by SDS–PAGE and immunoblotting.

# Data Availability

This study did not generate any unique datasets or code.

# Supplementary Information

# Acknowledgements

We thank Petra Robisch for the expert technical assistance with the "old50split." We thank Dr. Kai Hell for careful reading of the article and all the past and current members of the Mokranjac Group and Mito Club for the stimulating discussions and inspiring scientific ideas. We gratefully acknowledge the generous financial support from Deutsche Forschungsgemeinschaft to D Mokranjac (MO1944/3-1, MO1944/2-1, and NE101/28-1), Bayerisch-Tschechische Hochschulagentur to D Mokranjac and P Doležal (BTHA-JC-2022-33), The Ministry of Education, Youth and Sports of the

Czech Republic (LUABA22082) to P Doležal, Czech Science Foundation (20-25417S) to P Doležal. JSPS KAKENHI to T Endo (2222703, 15H05705, and 20H05689), JST CREST to T Endo (JPMJCR12M1), AMED CREST to T Endo (22gm1410002h0002).

## Author Contributions

MG Genge: conceptualization, investigation, visualization, and writing—original draft.

S Roy Chowdhury: investigation.

V Dohnálek: investigation, visualization, and writing—review and editing.

K Yunoki: investigation, visualization, and writing—review and editing.

T Hirashima: conceptualization, supervision, funding acquisition, investigation, visualization, project administration, and writing—review and editing.

T Endo: conceptualization, supervision, funding acquisition, visualization, project administration, and writing—review and editing.

P Doležal: conceptualization, supervision, funding acquisition, investigation, visualization, project administration, and writing—original draft, review, and editing.

D Mokranjac: conceptualization, supervision, funding acquisition, investigation, visualization, writing—original draft and project administration.

## Conflict of Interest Statement

The authors declare that they have no conflict of interest.

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
