## [Reviewer comments · Life Science Alliance]

Life Science Alliance

Two domains of Tim50 coordinate translocation of proteins across the two mitochondrial membranes

Marcel Gilbert Genge, Shalini Roy Chowdhury, Vit Dohnalek, Kaori Yunoki, Takashi Hirashima, Toshiya Endo, Pavel Dolezal and Dejana Mokranjac

DOI: <https://doi.org/10.26508/lsa.202302122>

Corresponding author(s): Dr. Dejana Mokranjac (LMU Munich)

Review Timeline:

Submission Date:	2023-04-28
Editorial Decision:	2023-06-12
Revision Received:	2023-08-28
Editorial Decision:	2023-09-11
Revision Received:	2023-09-13
Accepted:	2023-09-14

Scientific Editor: Dr. Eric Sawey, PhD

Transaction Report:

June 12, 2023

Re: Life Science Alliance manuscript #LSA-2023-02122-T

Dr. Dejana Mokranjac
LMU Munich
Biozentrum-Cell Biology
Großhadernerstr. 2
Planegg-Martinsried, Bavaria 82152
Germany

Dear Dr. Mokranjac,

Thank you for submitting your manuscript entitled "Two domains of Tim50 coordinate translocation of proteins across the two mitochondrial membranes" to Life Science Alliance. The manuscript was assessed by expert reviewers, whose comments are appended to this letter. We invite you to submit a revised manuscript addressing the Reviewer comments.

Thank you for this interesting contribution to Life Science Alliance. We are looking forward to receiving your revised manuscript.

Sincerely,

Eric Sawey, PhD
Executive Editor
Life Science Alliance
<http://www.lsa-journal.org>

B. MANUSCRIPT ORGANIZATION AND FORMATTING:

Reviewer #1 (Comments to the Authors (Required)):

Mitochondria consist of hundreds of proteins that are imported from the cytosol through translocases in the outer and inner mitochondrial membrane which are referred to as the TOM and TIM translocase. The constituents of these two complexes are long known, but how the TOM and TIM complex interact during preprotein translocation is only poorly understood. The inner membrane protein Tim50 plays a crucial role in this interplay. Tim50 is part of the TIM23 translocase and exposes a large domain into the IMS. Tim23 serves as a receptor during translocation and, in addition, contributes to the interaction of the TIM23 complex with IMS-exposed regions of the TOM complex. The IMS domain of Tim50 consists of two structurally defined domains called the core and the presequence-binding domain (PBD). In this study, the authors expressed both domains separately, the Core domain as part of a truncated Tim50 protein and the PBD as fusion with an IMS-targeting presequence. They convincingly show that this Tim50split protein is functional and supports protein import, although at reduced kinetics. The study reveals that both regions can be structurally separated, an observation, from which the authors draw a number of conclusions in the discussion. The study is of high technical quality, the text reads very well and necessary controls are presented. The study would be more compelling if the authors would provide some more data that shed light on the mechanism by which the two domains of Tim50 mediate protein import. At least one additional dataset should be added.

Major point

1. In all experiments shown, the authors used cells which express both domains of Tim50 simultaneously. Therefore, the only conclusion that can then be drawn is that Tim50 remains functional, even if the domains are disjointed which is not of high relevance for a general readership. The authors should generate mutants in which either of the parts is depleted individually and only one fragment is present, e.g. by GAL repression, even if similar experiments had been already performed before. With these two mutants (Core-down and PDB-down) they should perform (A) import experiments to test whether proteins can be transferred through the TIM to the IMS or even the matrix and (B) X-linking experiments to test whether the presence of the PDB is crucial for the transfer of precursors from the TOM to the Core domain or vice versa. A comprehensive data set on this needs to be added.

Minor points

2. Figures 1 and 2 are very data-poor. They should better be merged.
3. Figure S3B-E is technically sophisticated. However, the results do not add anything to this study. The authors should remove it. Conclusions can hardly be drawn without further controls. The claim that the PDB forms a dimer is not convincing as it might also be a receptor-substrate interaction at this stage.
4. Are the different Tim50 constructs well behaved and soluble? The authors make conclusions from Fig S3 on

solubility, but rather could test this directly with their yeast mutants. This would be more convincing.

5. The linker experiment is interesting but difficult to interpret. Whether it is really the flexibility of the linker which makes the difference here is not clear. The stiffer linkers with their large number of charges might also destabilize the structure of one of the Tim50 domains. Is it possible to use AlphaFold to predict the structures (and stabilities) of the different Tim50 mutants? This might be a nice addition to the supplement.

6. The discussion is extremely long, speculative and verbose. It would be considerably shortened and data should not be presented in the discussion part.

Reviewer #2 (Comments to the Authors (Required)):

Tim50 is an essential subunit of the TIM23 complex, which promotes protein import into the mitochondrial matrix or inner membrane. Tim50 exposes a soluble domain towards the intermembrane space to mediate substrate recognition and interaction with the TOM complex. This soluble part of Tim50 can be divided into the core domain and the peptide binding domain (PBD). Genge and colleagues demonstrate in an elegant study that both domains are essential for the function of Tim50. Excitingly, a yeast strains that express both domains separately are viable. The authors characterized this interesting mutant with genetic and biochemical studies. Thereby, they could show that the soluble domain of Tim50 can be functionally divided into two domains that are both important for protein import, particularly for the transfer from TOM to TIM23 complexes.

The reported findings are very interesting and the presented data are of high quality. I have a few minor suggestions and questions to improve the manuscript.

The authors propose that the PBD domain of Tim50 could confer stability to the proteins. While this idea is reasonable, the authors should provide some experimental data to analyze the stability of Tim50. E.g. does overexpression or downregulation of Tim50-PBD in the Split-Tim50 mutant affects growth or stability of Tim50?

In the presented data, it remains unclear how the expression of the PBD-domain of Tim50 supports growth in the Tim50 Split mutant. One possibility is that Tim50-PBD promotes protein transport already at the TOM complex. To this test idea, the authors may express a C-terminally fusion construct of Tom22 with Tim50-PBD and analyze whether it rescues the growth as well?

The labelling in Figure 4I is not clear and should be improved.

Dear Dr Sawey,

We would like to thank you and the two anonymous Reviewers for the careful evaluation, constructive criticism and thoughtful comments on our manuscript. The suggestions and comments of the Reviewers certainly helped to clarify several points in the manuscript and have significantly contributed to its improvement. Below we address point-by-point the various comments and suggestions raised in the Decision letter.

Reviewer #1 (Comments to the Authors (Required)):

Mitochondria consist of hundreds of proteins that are imported from the cytosol through translocases in the outer and inner mitochondrial membrane which are referred to as the TOM and TIM translocase. The constituents of these two complexes are long known, but how the TOM and TIM complex interact during preprotein translocation is only poorly understood. The inner membrane protein Tim50 plays a crucial role in this interplay. Tim50 is part of the TIM23 translocase and exposes a large domain into the IMS. Tim23 serves as a receptor during translocation and, in addition, contributes to the interaction of the TIM23 complex with IMS-exposed regions of the TOM complex. The IMS domain of Tim50 consists of two structurally defined domains called the core and the presequence-binding domain (PBD). In this study, the authors expressed both domains separately, the Core domain as part of a truncated Tim50 protein and the PBD as fusion with an IMS-targeting presequence. They convincingly show that this Tim50split protein is functional and supports protein import, although at reduced kinetics. The study reveals that both regions can be structurally separated, an observation, from which the authors draw a number of conclusions in the discussion. The study is of high technical quality, the text reads very well and necessary controls are presented. The study would be more compelling if the authors would provide some more data that shed light on the mechanism by which the two domains of Tim50 mediate protein import. At least one additional dataset should be added.

We thank the reviewer for his/her positive evaluation!

Major point

1. In all experiments shown, the authors used cells which express both domains of Tim50 simultaneously. Therefore, the only conclusion that can then be drawn is that Tim50 remains functional, even if the domains are disjointed which is not of high relevance for a general readership. The authors should generate mutants in which either of the parts is depleted individually and only one fragment is present, e.g. by GAL repression, even if similar experiments had been already performed before. With these two mutants (Core-down and PDB-down) they should perform (A) import experiments to test whether proteins can be transferred through the TIM to the IMS or even the matrix and (B) X-linking experiments to test whether the presence of the PBD is crucial for the transfer of precursors from the TOM to the Core domain or vice versa. A comprehensive data set on this needs to be added.

Following Reviewer's suggestion, we generated yeast centromeric plasmids in which the individual domains of Tim50 as well as the full-length protein were placed under the control of the GAL promoter and transformed them into the Tim50 shuffling strain in order to obtain yeast strains in which one of the two domains can be individually depleted. 50FL and 50split cells in which expression of Tim50 constructs was driven by the endogenous *TIM50* promoter were viable on both glucose- and galactose-containing 5-FOA medium (Figure for the Reviewers 1). In contrast, cells expressing full-length Tim50 from the GAL promoter were viable only on the galactose containing medium, in agreement with the previous results (Yamamoto et al, Cell, 2002, Geissler et al, Cell, 2002, Mokranjac et al, EMBO J, 2003). 50split PBD-down mutant was similarly only viable on galactose-containing medium. Unexpectedly, 50split Core-down cells were inviable on both glucose- and galactose-

containing medium (Figure for the Reviewers 1). We were therefore unfortunately not able to perform the suggested experiments with all the necessary controls. It should also be noted that the strain in which the full-length Tim50 was expressed from the centromeric plasmid under the *GAL* promoter was viable, however, produced considerably smaller colonies on 5-FOA/Galactose plates compared to the strain in which Tim50 was expressed from its endogenous promoter (Figure for the Reviewers 1). We speculate that the reason why the 50split cells, in which the core domain is expressed from the *GAL* promoter, are inviable and the ones that express plasmid-borne full-length Tim50 from the *GAL* promoter grow poorly on 5-FOA plates may be due to the previously observed detrimental effects of overexpression of laterally sorted TIM23 substrates for yeast cells (Weidberg and Amon, 2018, Science). The results presented in the 2018 Science paper showed that the overexpression of Tim50 is particularly problematic, compared to for example overexpression of cytochrome *b*₂. It is possible that overexpression of the core domain alone may be even more problematic for yeast cells, possibly due to the absence of PBD.

We cannot agree with the statement of the Reviewer that “the only conclusion that can be drawn (from cells expressing both domains simultaneously) is that Tim50 remains functional, even if the domains are disjointed which is not of high relevance for a general readership”. The results presented in this manuscript demonstrate that it was possible to address the functions of the individual Tim50 domains using 50split cells and isolated mitochondria. Even though both Tim50 domains are present in 50split mitochondria, our results clearly showed that only the core domain of Tim50 was efficiently recruited to the TIM23 complex, leading to the conclusion that it represents the main recruitment point of Tim50 to the TIM23 complex. Furthermore, using the same mitochondria, we unambiguously showed that it is the core domain of Tim50 that crosslinks to the incoming precursor proteins *in organello*, showing that it contains the main presequence-binding site. Thus, the 50split strain enabled us to clearly assign two previously defined functions of Tim50 to its core domain. If it were the PBD that was recruiting Tim50 to the TIM23 complex or was recognizing precursor proteins *in organello*, it would have been possible to see it. Also, experimental domain splitting offers a valuable approach to study function and interactions of proteins but it has not been widely used for membrane proteins and complexes and our study provides a good example as to how useful this approach can be in general.

Minor points

2. Figures 1 and 2 are very data-poor. They should better be merged.

The conclusion of Figure 1 is that both domains of Tim50 are essential for cell viability i.e. that neither of them is sufficient on its own, the finding that, after multiple publications showing that the deletion

of PBD alone is already lethal, many people may find unexpected. In contrast, Figure 2 describes our surprising finding that the function of the full-length Tim50 can be reconstituted *in vivo* from its individual domains expressed *in trans*. The two Figures thus address two completely different questions and lead to two completely different conclusions. For this reason we would prefer to keep the two Figures separated but would leave the final decision to the Editor.

3. Figure S3B-E is technically sophisticated. However, the results do not add anything to this study. The authors should remove it. Conclusions can hardly be drawn without further controls. The claim that the PBD forms a dimer is not convincing as it might also be a receptor-substrate interaction at this stage.

We removed Figures S3B and C, following Reviewer's recommendation. We however cannot agree with the statement that the results shown in Fig S3 do not add anything to this study. Presented data provide further clues as to what the possible function of the PBD could be. We find this very important as we were not able to recapitulate, in our *in organello* experiments shown here, previously published *in vitro* work which assigned the presequence-binding function to the PBD (Schulz et al, JCB; Lytovchenko et al, EMBO J).

[Figure removed by editorial staff per authors' request]

We agree with the Reviewer that crosslinking is able to capture transient interactions such as a receptor-substrate interaction. To address this possibility, we upscaled the crosslinking reaction, purified Tim50 and its crosslinking adducts by Ni-NTA affinity chromatography, separated the specifically bound material by SDS-PAGE and, after staining, excised the crosslinking product for MS analysis (Figure for the Reviewers 3, left panel). Peptide mass fingerprinting identified essentially only Tim50 in this band (Figure for the Reviewers 3, right panel), strongly suggesting that it represents a Tim50 dimer. If it were a mere receptor-substrate interaction, one would have expected to identify

many other TIM23 substrates on their way into mitochondria but, from ca 600 different possible substrates, not a single one was identified here by MS. Furthermore, we have previously shown that the time required to isolate mitochondria from yeast cells is sufficient to clear the TOM and TIM23 complexes of any translocating chains, unless they were artificially arrested (Popov-Celeketic et al, EMBO J, 2008).

We would leave the final decision to the Editor whether to move the remaining data to the Results section or remove it completely.

4. Are the different Tim50 constructs well behaved and soluble? The authors make conclusions from Fig S3 on solubility, but rather could test this directly with their yeast mutants. This would be more convincing.

We solubilized 50FL and 50split mitochondria with 1% digitonin and separated the solubilized from aggregated/non-solubilized material using ultracentrifugation (Figure for the Reviewers 4). Both the full-length protein and its two domains expressed *in trans* were predominantly found in the solubilized fraction. As it is not possible to distinguish whether the domains behave differently after recombinant expression and *in organello* or the PBD stabilized the core domain *in organello*, we removed the data on the recombinant expression (Figures S3B and C) and the corresponding text from the manuscript.

5. The linker experiment is interesting but difficult to interpret. Whether it is really the flexibility of the linker which makes the difference here is not clear. The stiffer linkers with their large number of charges might also destabilize the structure of one of the Tim50 domains. Is it possible to use AlphaFold to predict the structures (and stabilities) of the different Tim50 mutants? This might be a nice addition to the supplement.

We agree with the Reviewer that the linker experiments, and experiments with any mutant for that matter, may be difficult to interpret. This is the main reason why we tested two different flexible and three different rigid linkers of different chemical properties. As shown in Figure 7A, rigid linkers 1 and 3 are indeed enriched in charged residues, however, rigid linker 2 does not have a single charged residue in it. We thus think that the observed effects are not due to charge properties of the linkers but rather due to their stiffness.

To address the possibility raised by the Reviewer that the observed growth defects of the linker mutants are due to the potential instability of the resulting proteins, we analyzed the expression levels of the different Tim50 constructs. The levels of the various Tim50 linker mutants were indistinguishable from the wild type, showing that the introduction of the different linkers does not affect the stability of Tim50. This data is now included in the main Figures as Figure 7B and the text was modified accordingly.

We also followed Reviewers suggestion and used AlphaFold to predict the structures of the different Tim50 constructs (Figure for the

Reviewers 5). We observed no obvious differences between different Tim50 constructs, except in the linker regions. This was however to be expected as AlphaFold considers the entire structure and is known to perform rather poorly when it comes to predicting effects that small changes in the sequence have on the overall structure of a protein (<https://alphafold.ebi.ac.uk/fag> , <https://www.embl.org/news/science/alphafold-potential-impacts/>, Pak et al, 2023, PLOS One).

6. The discussion is extremely long, speculative and verbose. It would be considerably shortened and data should not be presented in the discussion part.

We removed data on the recombinant expression from Fig S3 and shortened the Discussion. We would however like to add that in this manuscript, the Discussion not only places the newly obtained data in the context of the already available knowledge but also aims to resolve the long standing conundrum in the field - namely that a critical function of Tim50, recognition of presequences, has been previously assigned to the only evolutionary non-conserved domain of Tim50. Our results did confirm the essential nature of this domain in yeast, however we obtained no evidence for its direct involvement in recognition of presequences. Rather our results point to its involvement in TOM-TIM23 cooperation. We can only speculate why this function would be essential for Fungi and not for other eukaryotes but a certain degree of speculation is normally allowed, if not even expected, in Discussions.

Reviewer #2 (Comments to the Authors (Required)):

Tim50 is an essential subunit of the TIM23 complex, which promotes protein import into the mitochondrial matrix or inner membrane. Tim50 exposes a soluble domain towards the intermembrane space to mediate substrate recognition and interaction with the TOM complex. This soluble part of Tim50 can be divided into the core domain and the peptide binding domain (PBD). Genge and colleagues demonstrate in an elegant study that both domains are essential for the function of Tim50. Excitingly, a yeast strains that express both domains separately are viable. The authors characterized this interesting mutant with genetic and biochemical studies. Thereby, they could show that the soluble domain of Tim50 can be functionally divided into two domains that are both important for protein import, particularly for the transfer from TOM to TIM23 complexes.

The reported findings are very interesting and the presented data are of high quality. I have a few minor suggestions and questions to improve the manuscript.

We thank the Reviewer for his/her positive evaluation!

The authors propose that the PBD domain of Tim50 could confer stability to the proteins. While this idea is reasonable, the authors should provide some experimental data to analyze the stability of Tim50. E.g. does overexpression or downregulation of Tim50-PBD in the Split-Tim50 mutant affects growth or stability of Tim50?

The 50split strain in which the expression of PBD can be regulated by the addition or omission of galactose was made as part of addressing comment 1 of Reviewer #1 (see Figure for the Reviewers 1). To address the point raised here, we grew the cells in galactose-containing medium and then spotted them on glucose-containing plates to abolish expression of PBD. Downregulation of PBD impaired growth of yeast cells (Figure for the Reviewers 6A), as expected from the lethal phenotype of the deletion of PBD. To analyze whether expression of PBD affected the stability of the core domain, cells were shifted for 24h from galactose- to glucose-containing medium, whole cell extracts were prepared and samples analyzed by SDS-PAGE and western blot. The levels of the core domain were indistinguishable between the two 50split strains even though the PBD expressed from the GAL promoter was nondetectable in glucose-containing medium (Figure for the Reviewers 6B). Therefore, absence of PBD does not seem to destabilize the core domain *in vivo*, at least not immediately.

When it comes to the effect of overexpression of PBD on growth of yeast cells, the results were less clear. We first analyzed the growth of the above mentioned 50split strain in which the expression of PBD is under the control of the GAL promoter in medium containing galactose. Overexpression of PBD in this case neither had an obvious effect on cell growth nor did it affect levels of the core domain (Figure for the Reviewers 6C and D). However, when we overexpressed PBD from the GPD promoter, the cell growth was clearly impaired, both on fermentable and on nonfermentable carbon source (Figure for the Reviewers 6E). Still, also in this case, the levels of the core domain remained unchanged (Figure for the Reviewers 6F).

Together with the experiments on the solubility of the individual domains in 50split mitochondria done in response to the Point 4 of Reviewer #1, these results provide no direct support for our hypothesis that PBD may directly affect the stability of the core domain. Therefore, we decided to remove the data on the recombinant expression and the corresponding text from the manuscript.

Figure for the Reviewers 6
 (A) Cells were grown in galactose containing-medium, 10-fold serial dilutions were made and spotted on glucose-containing medium. The plate was incubated at 30°C. (B) Cells were first grown in galactose-containing medium and then shifted for 24h to the medium containing glucose as the carbon source. Total cell extracts were made and analyzed by immunoblotting. (C) As in (A) except that the cells were spotted on galactose-containing medium. (D) As in (B) except that the cells were kept the whole time in galactose-containing medium. (E) Cells were grown in glucose-containing medium, serial dilutions were made and spotted on plates containing glucose and lactate as carbon sources, top and bottom panels, respectively. Plates were incubated at 30°C. (F) Whole cell extracts were made and analyzed by SDS-PAGE and immunoblotting.

[Figure removed by editorial staff per authors' request]

The labelling in Figure 4I is not clear and should be improved.

Crosslinks of Oxa1 precursor to both Tim50FL and the core domain were labelled with the same sign which is, we agree, confusing. We now used a hashtag (#) to indicate the crosslink to Tim50FL and a plus (+) to indicate the crosslink to the core domain.

September 11, 2023

RE: Life Science Alliance Manuscript #LSA-2023-02122-TR

Dr. Dejana Mokranjac
LMU Munich
Biozentrum-Cell Biology
Großhadernerstr. 2
Planegg-Martinsried, Bavaria 82152
Germany

Dear Dr. Mokranjac,

Thank you for submitting your revised manuscript entitled "Two domains of Tim50 coordinate translocation of proteins across the two mitochondrial membranes". We would be happy to publish your paper in Life Science Alliance pending final revisions necessary to meet our formatting guidelines.

- please add the Twitter handle of your host institute/organization as well as your own or/and one of the authors in our system
- please change the label of a supplementary table in its legend from Table ST1 to Table S1
- please add a callout for Table S1 to your main manuscript text;

To upload the final version of your manuscript, please log in to your account: <https://lsa.msubmit.net/cgi-bin/main.plex>

A. FINAL FILES:

B. MANUSCRIPT ORGANIZATION AND FORMATTING:

Sincerely,

Reviewer #1 (Comments to the Authors (Required)):

The authors satisfactorily addressed the points raised on the initial version. I now fully support the publication of this interesting study.

Reviewer #2 (Comments to the Authors (Required)):

The authors fully addressed my concerns. The presented data are of high quality and very interesting for the field. They provide new mechanistic insights about the function and the interplay of the two soluble domains of Tim50. I recommend publication of the manuscript in Life Science Alliance.

September 14, 2023

RE: Life Science Alliance Manuscript #LSA-2023-02122-TRR

Dr. Dejana Mokranjac
LMU Munich
Biozentrum-Cell Biology
Großhadernerstr. 2
Planegg-Martinsried, Bavaria 82152
Germany

Dear Dr. Mokranjac,

Thank you for submitting your Research Article entitled "Two domains of Tim50 coordinate translocation of proteins across the two mitochondrial membranes". It is a pleasure to let you know that your manuscript is now accepted for publication in Life Science Alliance. Congratulations on this interesting work.

Reviews, decision letters, and point-by-point responses associated with peer-review at Life Science Alliance will be published online, alongside the manuscript. We will be in touch with you to approve the content of these files prior to publication.

DISTRIBUTION OF MATERIALS:

Again, congratulations on a very nice paper. I hope you found the review process to be constructive and are pleased with how the manuscript was handled editorially. We look forward to future exciting submissions from your lab.

Sincerely,

Eric Sawey, PhD
Executive Editor
Life Science Alliance
<http://www.lsa-journal.org>